# Learning Graph Quantized Tokenizers

**Limei Wang,**[\*] **Kaveh Hassani**[\*]**, Si Zhang, Dongqi Fu, Baichuan Yuan, Weilin Cong, Zhigang Hua, Hao Wu, Ning Yao, Bo Long**

Meta AI

## Abstract

Transformers serve as the backbone architectures of Foundational Models, where domain-specific tokenizers allow them to adapt to various domains. Graph Transformers (GTs) have recently emerged as leading models in geometric deep learning, outperforming Graph Neural Networks (GNNs) in various graph learning tasks. However, the development of tokenizers for graphs has lagged behind other modalities, with existing approaches relying on heuristics or GNNs co-trained with Transformers. To address this, we introduce GQT (**G**raph **Q**uantized **T**okenizer), which decouples tokenizer training from Transformer training by leveraging multi-task graph self-supervised learning, yielding robust and generalizable graph tokens. Furthermore, the GQT utilizes Residual Vector Quantization (RVQ) to learn hierarchical discrete tokens, resulting in significantly reduced memory requirements and improved generalization capabilities. By combining the GQT with token modulation, a Transformer encoder achieves state-of-the-art performance on 20 out of 22 benchmarks, including large-scale homophilic and heterophilic datasets. The implementation is publicly available at https://github.com/limei0307/GQT.

## 1 Introduction

Following the success of Transformers (Vaswani et al., 2017) in Natural Language Processing (Devlin et al., 2019; Brown et al., 2020) and Computer Vision (Dosovitskiy et al., 2021), Graph Transformers (GTs) (Dwivedi & Bresson, 2020; Ying et al., 2021; Rampášek et al., 2022; Shirzad et al., 2023; Chen et al., 2023; Wu et al., 2022) have emerged as strong models in geometric deep learning. In contrast to message-passing Graph Neural Networks (GNNs), which are inherently constrained by strong locality inductive biases (Battaglia et al., 2018; Veličković et al., 2018; Hou et al., 2020; Hamilton et al., 2017; Kipf & Welling, 2017; Zheng et al., 2024), GTs exhibit greater expressivity due to their capacity to capture long-range interactions between nodes (Ma et al., 2023; Kim et al., 2022; Zopf, 2022). This is particularly beneficial in heterophilic settings where local alignment does not hold (Fu et al., 2024). This dichotomy highlights a fundamental trade-off between GNNs, which focus on local neighborhood aggregation, and GTs, which employ pairwise attention to model global graph structures. A natural question arises: can we synergistically integrate the strengths of both approaches to leverage the complementary benefits of local and global representations? Specifically, is it possible to harness the locality-aware representations learned by GNNs to construct discrete tokens, thereby enabling GTs to operate efficiently while still capturing salient graph properties?

GTs require consideration of both graph structure and features, as nodes with identical features will otherwise be projected into the same representation regardless of their surrounding structures (Hoang et al., 2024). There are three general approaches to address this limitation (Hoang et al., 2024): (1) node feature modulation, which involves injecting structural information into the node features; (2) context node sampling, where a sampling strategy is used to construct a sequence over the neighbor nodes; and (3) modifying the architecture of a vanilla Transformer to directly incorporate structural biases. Given that Transformers are universal approximators of sequence-to-sequence functions (Yun et al., 2020) and considering the rapid developments in efficient implementation of Multi-Head Attention (MHA) module (Dao et al., 2022; Liu et al., 2024), which enables longer context sizes (Reid et al., 2024), we argue that a well-designed graph tokenizer can allow a vanilla

---

[\*]Equal Contributions

Transformer to efficiently process even large-scale graphs. Recent studies on applying Large Language Models (LLMs) to Text-Attributed Graphs (TAGs) have shown surprisingly strong performance gains surpassing those of GNNs, suggesting that vanilla Transformers are indeed capable of effectively learning graph structures (Ye et al., 2024; Xu et al., 2025). Nonetheless, LLMs are not efficient at inference time. Our goal is to devise a lightweight and efficient graph tokenizater that allows vanilla Transformer encoders to learn graph structures effectively.

Tokenizers typically employ self-supervised objectives to abstract data into a sequence of discrete tokens, allowing Transformers to learn representations across various modalities as a unified stream of data. The discretization is usually achieved through vector quantization techniques (Van Den Oord et al., 2017; Lee et al., 2022), which offer several benefits, including: (1) significantly reduced memory requirements, (2) improved inference efficiency, (3) allowing Transformers to focus on long-range dependencies rather than local information, and (4) the capacity to learn more high-level representations due to a compact latent space (Yuan et al., 2021; Yu et al., 2022a). These advantages are particularly important in auto-regressive generative modeling, where quantized tokens allow Transformers to generate high-quality outputs in multiple modalities (Dubey et al., 2024; Lee et al., 2022; Dhariwal et al., 2020; Ramesh et al., 2021; Team, 2024). Despite its importance in other domains, tokenization remains under-explored for graph-structured data. To address this limitation, we propose the **Graph Quantized Tokenizer (GQT)**, a novel approach that learns a hierarchical sequence of tokens over graphs using self-supervised objectives tailored to graph-structured data. More specifically, our contributions are as follows:

- We propose a graph tokenizer that uses multi-task graph self-supervised learning to train a graph encoder, enabling it to fully capture local interactions and allowing the Transformer to focus on long-range dependencies.

- Our approach adapts Residual Vector Quantization (RVQ) within the graph tokenizer to learn hierarchical discrete tokens, resulting in significantly reduced memory requirements and improved generalization capabilities.

- We introduce a novel combination of semantic edges and random walks to facilitate access to long-range interactions, and employ hierarchical encoding and gating mechanisms to modulate the tokens and provide informative representations to the Transformer.

- Through extensive experiments on both homophilic and heterophilic datasets, including large-scale and long-range benchmarks, we demonstrate that our tokenizer enables Transformer encoders to achieve state-of-the-art performance on 20 out of 22 benchmarks while substantially reducing the memory footprint of the embeddings.

## 2    RELATED WORKS

**Graph Transformers (GTs)** have shown promising performance on various graph learning tasks, surpassing GNNs on many benchmarks. Designing GTs can be broadly categorized into two directions (Hoang et al., 2024; Müller et al., 2024): (1) modifying the vanilla Transformer architecture to incorporate structural inductive biases, or (2) encoding the input graph to make it compatible with the vanilla Transformer. Early examples of the first approach include Graph Attention Network (GAT) (Veličković et al., 2018), which uses an attention module to compute pairwise node attention and masks the attention matrix based on connectivity information. Subsequent works have replaced the scaled-dot attention module with various structure-aware sparse attention modules (Rampášek et al., 2022; Bo et al., 2023; Ying et al., 2021; Deng et al., 2024; Wu et al., 2023b; Liu et al., 2023a; Chen et al., 2022; Dwivedi & Bresson, 2020; Shirzad et al., 2023; Ma et al., 2023). Graph Memory Network (GMN) (Khasahmadi et al., 2020) is an example of the second approach, which passes non-linear projections of node features and structural encoding to a Transformer-like model. Structural encodings such as Laplacian eigenvectors or Random walk-based encoding (Dwivedi et al., 2022a; Ma et al., 2023; Cantürk et al., 2024), allow injecting structural information directly into the node features. Some works use GNNs to encode local structure along with node features into embeddings that are passed to vanilla Transformers to capture long-range dependencies (Rong et al., 2020; Wu et al., 2021; Chen et al., 2023; 2022). Recent studies leverage LLMs, where graphs are represented through natural language, and an LLM performs graph-related tasks through in-context learning, instruction-tuning, or soft-prompting (Fatemi et al., 2024; Ye et al., 2024; He et al., 2024). For a detailed survey on GTs, see (Müller et al., 2024; Hoang et al., 2024).

**Graph Tokenization** provides GTs with rich node tokens that encapsulate both structural and semantic information. TokenGT (Kim et al., 2022) treats nodes and edges as independent tokens defined by their features, type identifiers, and structural encodings. NAGphormer (Chen et al., 2023) represents each node as a set of $L$ tokens, where the $l^{th}$ token is the representation of the node from the $l^{th}$ hop aggregation. In contrast, GraphiT (Mialon et al., 2021) defines a node token as the concatenation of its feature and representation from a graph convolutional kernel network (GCKN). VCR-Graphormer (Fu et al., 2024) expands the notion of node tokens to include sequences comprising the node feature and features of semantically and community-related neighboring nodes. SGT (Liu et al., 2023c) is a non-parametric tokenizer designed for molecular tasks, which unlike motif-based tokenizers (Zhang et al., 2021; Jin et al., 2018) or GNN pre-training methods (Xia et al., 2023), simplifies the tokenization process to a non-parametric graph operator without non-linearity. NodePiece (Galkin et al., 2022) is a knowledge-graph tokenizer that represents a target node as a hash of its top-k closest anchors, their distances, and relational context. While Vector Quantization (VQ) has been explored in other modalities (Van Den Oord et al., 2017; Lee et al., 2022; Yu et al., 2022b; Van Kempen et al., 2024; Li et al., 2024), its application in graph learning is limited. Notable exceptions include VQ-GNN (Ding et al., 2021), which uses quantized representations combined with a low-rank version of the graph convolution matrix to avoid neighbor explosion problem, VQGraph (Yang et al., 2024), which employs VQ for distilling a GNN into an MLP, and NID (Luo et al., 2024a), which uses VQ to learn discrete node IDs for downstream prediction tasks.

## 3 PRELIMINARIES

**Messag-Passing GNNs**. Let $\mathcal{G}$ denote the space of graphs. A graph $g \in \mathcal{G}$ is defined as $(\mathcal{V}, \mathcal{E}, \mathbf{X}, \mathbf{E})$ where $\mathcal{V}$ is the set of nodes and $\mathcal{E} \subseteq \mathcal{V} \times \mathcal{V}$ is the set of edges. $\mathbf{X} \in \mathbb{R}^{|\mathcal{V}| \times d_x}$ represents the node features of dimension $d_x$, and $\mathbf{E} \in \mathbb{R}^{|\mathcal{V}| \times |\mathcal{V}| \times d_e}$ represents the edge features of dimension $d_e$. A message-passing GNN takes $g$ as input and learns representations $h_v^l$ for $v \in \mathcal{V}$ ($h_v^0 = x_v$) in each layer $l$ as follows (Gilmer et al., 2017):

$$h_v^l = f_\theta^l \left( h_v^{l-1}, g_\phi^l \left( \left\{ \left( h_v^{l-1}, h_u^{l-1}, e_{uv} \right) | u \in \mathcal{N}_v \right\} \right) \right) \tag{1}$$

where $f_\theta$ and $g_\phi$ are known as combine and aggregate functions, respectively. $\mathcal{N}_v$ denotes the set of immediate neighbors of the node $v$. Once the node representations are computed, we can perform various tasks including node classification as $\text{MLP}(h_v)$, edge prediction as $\text{MLP}(h_u \odot h_v)$, or graph classification as $\text{MLP}(\mathcal{R}(\{h_u | u \in \mathcal{V}\}))$, where $\mathcal{R}$ is a pooling (readout) function.

**Graph Transformers** use a tokenizer $T_v = \mathcal{T}_\psi(\mathcal{N}(v))$ to map each node $v \in \mathcal{V}$ into a sequence of tokens $T_v$ by considering a notion of neighborhood $\mathcal{N}$. The simplest design is when $\mathcal{N}$ is zero-hop neighborhood (i.e., the node itself) and $\mathcal{T}_\psi$ is a node feature lookup function. The neighborhood $\mathcal{N}$ can be extended to include the node's ego network (Zhao et al., 2021) or top-k Random Walk based neighbors (Fu et al., 2024), and $\mathcal{T}_\psi$ can be enhanced to representations from a GNN (Chen et al., 2023). Node tokens along with positional encodings (PE) are passed to the Transformer as $h_v^0 = [T_v || \text{PE}(v)]$. The representations in the $l^{th}$ layer of a Transformer encoder are computed as:

$$h_v^l = \text{LN}\left(\text{MHA}\left(\text{LN}\left(h_v^{l-1}\right)\right) + h_v^{l-1}\right) \tag{2}$$

$$h_v^l = h_v^l + \text{MLP}\left(h_v^l\right) \tag{3}$$

where LN and MHA are LayerNorm and multi-head attention, respectively. Similar to Transformer encoders in other modalities (Devlin et al., 2019; Dosovitskiy et al., 2021), we can append a special classification token ([CLS]) to the input and use its representation to perform various classification tasks on the graph: $\text{MLP}\left(h_{[\text{CLS}]}\right)$.

**Vector Quantization** projects embeddings $\mathbf{X} \in \mathbb{R}^{n \times d_x}$ into a more compact space of codebooks $\mathbf{C} \in \mathbb{R}^{k \times d_c}$, where $k \ll n$. The codebooks can be learned by minimizing various objectives such as K-means clustering. The new representation of $x_i$ is then computed as (Van Den Oord et al., 2017):

$$z(x_i) = c_k \quad \text{where} \quad k = \arg\min_j \|x_i - c_j\|_2^2 \tag{4}$$

Building upon this concept, Residual-VQ (RVQ) (Lee et al., 2022) extends VQ to a sequence of codebooks, where each consecutive codebook quantizes the residual error from the previous codebook, i.e., $r_i = z_i - c_k$. This hierarchical approach constructs a multi-level quantized representation, enhancing the overall quantization quality. More details of RVQ are included in Appendix B.

## 4 SELF-SUPERVISED GRAPH TOKENIZATION

### 4.1 TOKENIZER PROPERTIES

Our goal is to design a graph tokenizer that learns node tokens that exhibit three key characteristics:

**Modeling Local Interactions**. The tokens should encapsulate local interactions, allowing the Transformer to focus on long-range dependencies. This is analogous to Vision Transformers (ViTs), where the Transformer attends to image patches instead of pixels (Dosovitskiy et al., 2021; Liu et al., 2021). To achieve this, we leverage GNNs as the tokenizer encoder to model local interactions in the representation space (Battaglia et al., 2018). Our design accommodates various GNN layer choices without constraints; for simplicity, we opt for a GAT encoder (Veličković et al., 2018).

**Memory Efficiency**. The tokens also should be compact to facilitate efficient memory usage. To achieve this, we introduce a Residual-VQ (RVQ) (Lee et al., 2022) layer to quantize the GNN representations into a sequence of discrete tokens. Quantization not only helps with generalization due to its regularization effect but also significantly reduces memory usage. Using an RVQ with $c$ codebooks (typically $c = \{2, \cdots, 8\}$), a graph with feature matrix $\mathbf{X} \in \mathbb{R}^{N \times d_x}$ can be represented as $\mathbf{X}_Q \in \mathbb{N}^{N \times c}$ and codebook representation of $\mathbf{C} \in \mathbb{R}^{c \times K \times d_c}$, where $c$ is the number of codebooks (i.e., levels of quantization), $K$ is the codebook size, and $d_c$ is the code dimension. To illustrate the benefits of this approach, consider a graph with $10^6$ nodes and a feature dimension of 1024 ($\mathbf{X} \in \mathbb{R}^{10^6 \times 1024}$). Using an RVQ with 3 codebooks and a codebook size of 256, this graph can be represented as $\mathbf{X}_Q \in \mathbb{N}^{10^6 \times 3}$ plus $\mathbf{C} \in \mathbb{R}^{3 \times 256 \times 1024}$, resulting in a 270-fold reduction in memory.

**Robustness and Generalization**. The tokens should be robust and generalizable. To achieve this, we rely on graph self-supervised learning. Self-supervised representations have been shown to be more robust to class imbalance (Liu et al., 2022) and distribution shift (Shi et al., 2023), while also capturing better semantic information (Assran et al., 2023) compared to representations learned through supervised objectives. Moreover, self-supervised graph representations have demonstrated superior performance on downstream tasks compared to representations learned in a fully supervised manner, indicating better generalization capabilities (Hu et al., 2020b; Sun et al., 2020; You et al., 2020; Fu et al., 2020; You et al., 2021; Hassani & Khasahmadi, 2020; Veličković et al., 2019; Zhu et al., 2020b). Additionally, multi-task learning with self-supervised objectives has been shown to achieve better performance on downstream tasks (Doersch & Zisserman, 2017; Ghiasi et al., 2021). To leverage these benefits, we propose training the GNN encoder with three self-supervised objectives. Unlike RQ-VAE (Lee et al., 2022), which uses reconstruction as its primary objective, we employ graph-specific objectives to capture the nuances of both structure and features.

### 4.2 TRAINING

To capture different aspects of information, we employ a multi-task learning framework that leverages three distinct families of graph self-supervised objectives: student-teacher distillation (Thakoor et al., 2022), masked autoencoding (Hou et al., 2022), and Infomax (Veličković et al., 2019). We also introduce a commitment loss (Van Den Oord et al., 2017) to enforce alignment between learned node representations and the codebook representations. Specifically, the GNN encoder is trained through gradient descent to minimize a loss function comprising of three terms, where $\beta$ is the loss weight:

$$\mathcal{L} = \mathcal{L}_{\text{dgi}} + \mathcal{L}_{\text{gmae2}} + \beta \mathcal{L}_{\text{commit}} \tag{5}$$

The first term is the Deep Graph Infomax (DGI) (Veličković et al., 2019) objective, which maximizes mutual information (MI) between node representations and graph (sub-graph) representations, based on the Jensen-Shannon divergence between the joint and product of marginals as follows:

$$\mathcal{L}_{\text{dgi}} = \mathbb{E}\left(\sum_{v \in g} \log\left(\mathcal{D}\left(h_v, h_g\right)\right) + \sum_{u \in \tilde{g}} \log\left(1 - \mathcal{D}\left(h_u, h_g\right)\right)\right) \tag{6}$$

where $h_u$ is the representation of node $u$. $h_g$ is the global (sub-graph/graph) representation, computed as the mean of node representations. $\tilde{g}$ is the corrupted version of the original graph, with the same structure but randomly shuffled features, providing negative examples for contrastive learning. $\mathcal{D}\left(h_u, h_g\right) = \sigma\left(h_u^T \mathbf{W} h_g\right)$ is the discriminator that scores whether a node belongs to the graph, and is defined as a bilinear classifier.

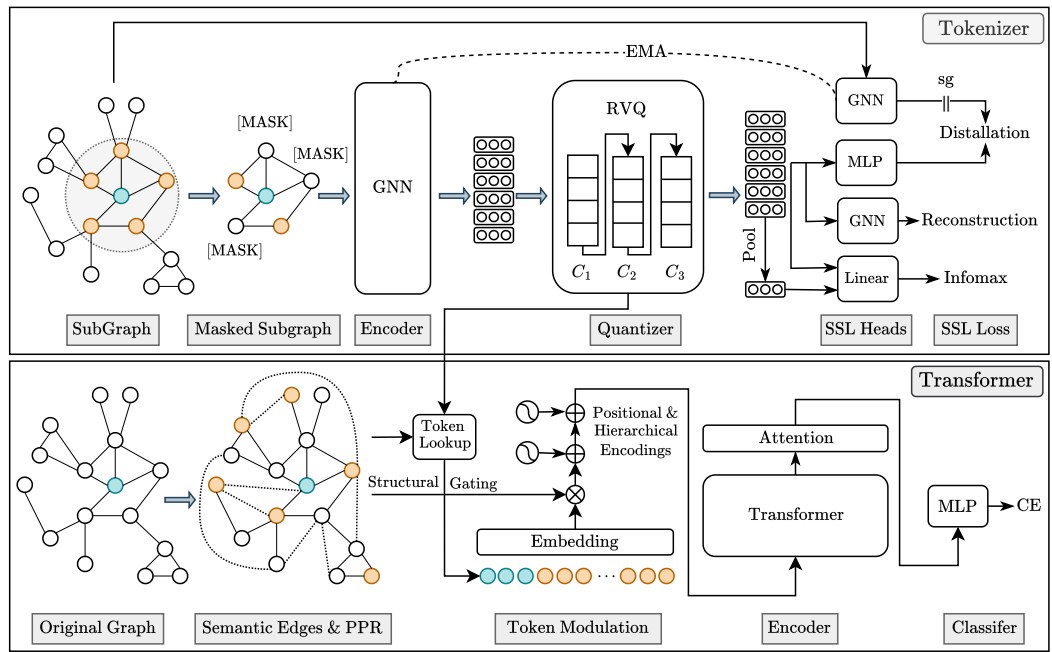

Figure 1: Overview of our proposed Graph Quantization Transformer (GQT) consisting of three main components: (1) a GNN to encode local interactions, (2) vector quantization for compact representation, and (3) generative and contrastive heads for robust representation learning. We also utilize a Transformers encoder to model long-range interactions. We augment the graph with semantic edges (dashed lines) and generate a sequence for each node based on Personalized PageRank scores. We then modulate the tokens through hierarchical encoding and structural gating, and feed them into the Transformer and aggregate the learned representations through an attention module before passing it to the classification head.

The second term is the GraphMAE2 objective (Hou et al., 2023), which combines the generative loss of GraphMAE (Hou et al., 2022) with the teacher-(noisy)student distillation loss of BGRL (Thakoor et al., 2022). This combination enables the model to avoid overfitting and learn more semantic representations. The GraphMAE2 loss is computed as follows:

$$\mathcal{L}_{\text{gmae2}} = \sum_{v \in \tilde{g}} \left( 1 - \frac{x_v^T . \tilde{h}_v}{\|x_v^T\| . \|\tilde{h}_v\|} \right)^\gamma + \lambda \sum_{v \in g} \left( 1 - \frac{h_v^T . \tilde{h}_v}{\|h_v^T\| . \|\tilde{h}_v\|} \right)^\gamma \tag{7}$$

where $\tilde{g}$ is the masked graph, $\tilde{h}_v$ is the node representation of a masked node learned by the noisy student, $h_v$ is the corresponding node representation learned by the teacher over the original graph, and $\gamma \geq 1$ is a scaling factor. The teacher's parameters are updated using an Exponential Moving Average (EMA) of the noisy student's parameters.

The third term is the commitment loss, which encourages the representations to get close to their corresponding codebook embeddings within the RVQ layer. This loss is computed as:

$$\mathcal{L}_{\text{commit}} = \frac{1}{|\mathcal{V}|} \sum_{v \in g} \|h_v - \text{sg}\,[c_k]\,\|_2 \tag{8}$$

where sg is the stop-gradient operator, and $c_k$ is the representation of the codebook that $h_v$ is assigned to (i.e., the centroid or prototype vector). Note that this loss only affects the node representations and does not update the codebooks.

To initialize and update the codebooks, we employ K-Means clustering and EMA with weight decay $\tau \in [0, 1]$, respectively. Specifically, the codebooks are updated as follows:

$$c_k^t = \tau c_k^{t-1} + (1 - \tau) \frac{1}{|\mathcal{V}_k|} \sum_{v \in \mathcal{V}_k} h_v \tag{9}$$

where $\mathcal{V}_k$ is the set of nodes assigned to codebook $c_k$. This update rule allows the codebooks to adapt to the changing node representations while maintaining stability.

## 5 GRAPH TRANSFORMER

### 5.1 GRAPH SERIALIZATION

Once the tokenizer is trained, each node $v \in \mathcal{V}$ is mapped to $c$ discrete tokens: $T_v = [t_1^v, \cdots, t_c^v] \in \mathbb{N}^c$ (i.e., $T_v = \mathbf{X}_Q[v]$), encoding local interactions of that node. We then need to serialize the graph in order to input it to the Transformer.

**Semantic Edges**. To enable the Transformer to capture long-range interactions, the input should consist of a sequence of tokens from nodes that are likely to have long-range dependencies. To facilitate this, we first augment the graph with *semantic edges* denoted as $\mathcal{E}_s$, which are computed as follows:

$$\mathcal{E}_s = \left\{ e_{u,v} \mid \underset{u \in \mathcal{V}}{\arg \operatorname{topk}} \operatorname{sim}\left(f\left(x_u\right), f\left(x_v\right)\right) \forall v \in \mathcal{V} \right\} \tag{10}$$

where $\operatorname{sim}(\cdot, \cdot)$ denotes the similarity function, $x_u$ is the feature vector of node $u$, and $f$ is a projection function. We use cosine similarity as the similarity function and Principal Component Analysis (PCA) as the projection function. The semantic edge augmentation effectively creates sparse edges between each node and its k-nearest neighbors in the feature space, enhancing the model's ability to recognize and utilize long-range dependencies.

**Structural Serialization**. We combine the semantic edges with the original edges and use Personalized PageRank (PPR) to generate a sequence per target node. This enriches the sequence with information beyond local interactions, allowing the Transformer to access potential long-range dependencies. We construct the sequence $S_v$ for each node $v$ as follows:

$$S_v = \left[ T_v \| T_u \|_{u \in \arg \operatorname{topk} \operatorname{PPR}(v, \mathcal{E} \cup \mathcal{E}_s)} \right] \tag{11}$$

where $S_v = [t_1^v \cdots t_c^v \mid t_1^{u_1} \cdots t_c^{u_1} \mid \cdots \mid t_1^{u_k} \cdots t_c^{u_k}]$ is a sequence of length $c \times (k + 1)$, comprising discrete tokens that represent the target node $v$, followed by discrete tokens of the top-$k$ relevant nodes to node $v$. These relevant nodes are determined based on PPR scores. Note that the computation of semantic edges and PPR sequences is performed only once as a pre-processing step, thereby reducing the computational overhead during training.

### 5.2 TOKEN MODULATION

**Token Embeddings**. There are $c \times K$ possible discrete tokens, where $c$ is the number of codebooks and $K$ is the codebook size. We randomly initialize an embedding matrix $\mathbf{X}_T \in \mathbb{R}^{c \times K \times d_x}$, which is trained end-to-end alongside the Transformer. To further enrich the token representations, we introduce an additional token for each node that aggregating the embeddings of its assigned codebooks from the pretrained tokenizer:

$$h_c^v = \sum_{i=1}^c \mathbf{C}[i, t_i^v] \tag{12}$$

where $\mathbf{C}[i, j]$ is the embedding corresponding to index j in the ith codebook. We found that adding this explicit aggregated token from the codebook leads to better performance compared to initializing $\mathbf{X}_T$ directly with $\mathbf{C}$. The input representation of the sequence for node $v$ is then defined as:

$$S_v = \left[ \mathbf{X}_T[i, t_i^v] \overset{c}{\underset{i=1}{\|}} h_c^v \| \mathbf{X}_T[i, t_i^{u_1}] \overset{c}{\underset{i=1}{\|}} h_c^{u_1} \| \cdots \| \mathbf{X}_T[i, t_i^{u_k}] \overset{c}{\underset{i=1}{\|}} h_c^{u_k} \right] \tag{13}$$

where $[\mathbf{X} \| \mathbf{Y}]$ denotes concatenation of sequences $\mathbf{X}$ and $\mathbf{Y}$. This representation combines the individual token embeddings with the aggregated codebook embeddings, providing a more comprehensive and nuanced input to the Transformer.

**Structural Gating**. In order to provide the Transformer with the global structural importance scores of the nodes within the sequence with respect to the target node, we introduce a gating mechanism over the input token embeddings as follows:

$$S_v = S_v \odot \operatorname{Softmax}\left(\operatorname{topk} \operatorname{PPR}\left(v, \mathcal{E} \cup \mathcal{E}_s\right)\right) \tag{14}$$

where we first apply a softmax function with temperature $\tau = 1$ to normalize the PPR scores, and then multiply each node token's representation by its corresponding normalized score.

**Positional Encoding**. We also introduce two trainable positional encodings to the input tokens. The first positional encoding enables the Transformer to distinguish between tokens from different nodes, while the second encoding, referred to as hierarchical encoding, allows the Transformer to recognize the hierarchy level of each token within the codebooks. We randomly initialize the positional encodings $\mathbf{PE} \in \mathbb{R}^{(k+1) \times d_x}$ and $\mathbf{HE} \in \mathbb{R}^{c \times d_x}$ and sum them with the encoding of their corresponding token. For example, the final encoding of the token $j$ of the node $i$ within the sequence is computed as: $x = \mathbf{X}_T[j, t_j^{u_i}] + \mathbf{PE}[i] + \mathbf{HE}[j]$. Note that we did not use any structural encoding, such as Laplacian eigenvectors, as we did not observe any significant gains from them.

## 5.3 TRANSFORMER ENCODER

We use $l$ layers of standard Transformer encoder with flash attention (Dao et al., 2022) to generate contextual representations per token in the sequence: $\mathbf{H}^{(l)} \in \mathbb{R}^{(c+1) \times (k+1) \times d_h}$. We then aggregate the token representations for $j$-th node in the sequence by summing along the token dimension:

$$\mathbf{H}_{v_j} = \sum_{i=1}^{c+1} \mathbf{H}^{(l)}[i, j] \in \mathbb{R}^{(k+1) \times d_h} \tag{15}$$

To obtain a single representation for the entire sequence, We further aggregate the representation using a linear attention layer:

$$h = \sum_{i=1}^{k+1} \alpha_i h_i \quad \text{where} \quad \alpha_i = \frac{\exp(\mathbf{W} h_i)}{\sum_j \exp(\mathbf{W} h_j)} \tag{16}$$

We feed the resulting representation into a fully-connected classifier and train the model end-to-end using cross-entropy loss. Note that during inference, only the Transformer and classifier are utilized, as the tokenizer is pretrained and the sequences are pre-computed. Furthermore, since we only require discrete tokens and codebook embeddings, our approach enables efficient memory usage, regardless of graph size, allowing for efficient training and inference on large-scale graphs.

## 6 EXPERIMENTS

We evaluate GQT on both medium- and large-scale graph learning tasks, encompassing 22 homophilic, heterophilic, and long-range benchmarks. We follow the established experimental protocols from previous works to ensure fair comparisons. Details of the datasets, experimental setup, and hyperparameters are provided in Appendices C and D, respectively.

### 6.1 COMPARISON WITH STATE-OF-THE-ART

**Long-Range Benchmarks**. We use four datasets from the Long-Range Graph Benchmark (LRGB) (Dwivedi et al., 2022b), including the Peptides-Func dataset for graph classification with Average Precision (AP) metric, the Peptides-Struct dataset for graph regression with Mean Absolute Error (MAE) metric, the COCO-SP dataset for inductive node classification with macro F1 metric, and the PCQM-Contact for link prediction with Mean Reciprocal Rank (MRR) metric. We compare our results to baselines reported in (Wang et al., 2024). The results shown in Table 1 suggest that GQT is able to capture long-range dependencies and performs well on various graph prediction tasks.

**Homophilic Node Classification.** We use eight medium-scale homophilic datasets including: Cora-Full (Bojchevski & Günnemann, 2017), CiteSeer, PubMed (Yang et al., 2016), Amazon Computers, Amazon Photos, Co-author CS, Co-author Physics (Shchur et al., 2018), and WikiCS (Mernyei & Cangea, 2020). We compare our results with eight GNNs including: GCN (Kipf & Welling, 2017), GAT, APPNP (Gasteiger et al., 2018), GPRGNN (Chien et al., 2020), GraphSAINT (Zeng et al., 2020), GraphSAGE (Hamilton et al., 2017), PPRGo (Bojchevski et al., 2020), and GTAND+ (Feng et al., 2022). We also compare against ten GTs including GT (Dwivedi & Bresson, 2020), Graphormer (Ying et al., 2021), SAN (Kreuzer et al., 2021), GraphGPS (Rampášek et al., 2022), GOAT (Kong et al., 2023), NodeFormer (Wu et al., 2022), DiffFormer (Wu et al., 2023a), NAGphormer (Chen et al., 2023), Exphormer (Shirzad et al., 2023), and VCR-Graphormer (Fu et al., 2024). The baseline performance is reported from (Wu et al., 2023b; Luo et al., 2024a). GQT outperforms the baseline

Table 1: Mean performance on inductive long-range benchmarks over five runs.

| Task | Graph Classification | Graph Regression | Node Classification | Link Prediction |
|---|---|---|---|---|
| Dataset | Peptides-Func | Peptides-Struct | COCO-SP | PCQM-Contact |
| #Graphs | 15,535 | 15,535 | 123,286 | 529,434 |
| Avg. #Nodes | 150.94 | 150.94 | 476.88 | 30.14 |
| Avg. #Edges | 307.30 | 307.30 | 2,693.67 | 61.09 |
| Metric | AP ↑ | MAE ↓ | F1 ↑ | MRR ↑ |
| GCN | 0.5930±0.0023 | 0.3496±0.0013 | 0.0841±0.0010 | 0.3234±0.0006 |
| Exphormer | 0.6258±0.0092 | 0.2512±0.0025 | 0.3430±0.0108 | **0.3587±0.0025** |
| GPS | 0.6535±0.0041 | 0.2500±0.0005 | 0.3412±0.0044 | 0.3337±0.0006 |
| Graph-Mamba | 0.6739±0.0087 | 0.2478±0.0016 | 0.3960±0.0175 | 0.3395±0.0013 |
| GQT (Ours) | **0.6903±0.0085** | **0.2452±0.0018** | **0.4007±0.0125** | 0.3427±0.0012 |

Table 2: Mean node classification accuracy on medium-scale homophilic datasets over five runs.

| | | CoraFull | CiteSeer | PubMed | Computer | Photo | CS | Physics | WikiCS |
|---|---|---|---|---|---|---|---|---|---|
| Dataset | #Nodes | 19,793 | 3,327 | 19,717 | 13,752 | 7,650 | 18,333 | 34,493 | 11,701 |
| | #Edges | 126,842 | 4,522 | 88,651 | 491,722 | 238,163 | 163,788 | 495,924 | 216,123 |
| | #Features | 8,710 | 3,703 | 500 | 767 | 745 | 6,805 | 8,415 | 300 |
| | #Classes | 70 | 6 | 3 | 10 | 8 | 15 | 5 | 10 |
| GNN | GCN | 61.76±0.14 | 76.50±1.36 | 86.54±0.12 | 89.65±0.52 | 92.70±0.20 | 92.92±0.12 | 96.18±0.07 | 77.47±0.85 |
| | GAT | 64.47±0.18 | 76.55±1.23 | 86.32±0.16 | 90.78±0.13 | 93.87±0.11 | 93.61±0.14 | 96.17±0.08 | 76.91±0.82 |
| | APPNP | 65.16±0.28 | 76.53±1.16 | 88.43±0.15 | 90.18±0.17 | 94.32±0.14 | 94.49±0.07 | 96.54±0.07 | 78.87±0.11 |
| | GPRGNN | 67.12±0.31 | 77.13±1.67 | 89.34±0.25 | 89.32±0.29 | 94.49±0.14 | 95.13±0.09 | 96.85±0.08 | 78.12±0.23 |
| | GraphSAINT | 67.85±0.21 | – | 88.96±0.16 | 90.22±0.15 | 91.72±0.13 | 94.41±0.09 | 96.43±0.05 | – |
| | GraphSAGE | – | 75.58±1.33 | 87.48±0.38 | 91.20±0.29 | 94.59±0.14 | 93.91±0.13 | 96.49±0.06 | 74.77±0.95 |
| | PPRGo | 63.54±0.25 | – | 87.38±0.11 | 88.69±0.21 | 93.61±0.12 | 92.52±0.15 | 95.51±0.08 | 78.12±0.23 |
| | GRAND+ | 71.37±0.11 | – | 88.64±0.09 | 88.74±0.11 | 94.75±0.12 | 93.92±0.08 | 96.47±0.04 | – |
| GT | GT | 61.05±0.38 | – | 88.79±0.12 | 91.18±0.17 | 94.74±0.13 | 94.64±0.13 | 97.05±0.05 | – |
| | Graphormer | OOM | – | OOM | OOM | 92.74±0.14 | 94.64±0.13 | OOM | – |
| | SAN | 59.01±0.34 | – | 88.22±0.15 | 89.93±0.16 | 94.86±0.10 | 94.51±0.15 | OOM | – |
| | GraphGPS | 55.76±0.23 | 76.99±1.12 | 88.94±0.16 | OOM | 95.06±0.13 | 93.93±0.15 | OOM | 78.66±0.49 |
| | GOAT | – | 76.89±1.19 | 86.87±0.24 | 90.96±0.90 | 92.96±1.48 | 94.21±0.38 | 96.24±0.24 | 77.00±0.77 |
| | NodeFormer | – | 76.33±0.59 | 89.32±0.25 | 86.98±0.62 | 93.46±0.35 | 95.64±0.22 | 96.45±0.28 | 74.73±0.94 |
| | DIFFormer | – | 76.72±0.68 | 89.51±0.67 | 91.99±0.76 | 95.10±0.47 | 94.78±0.20 | 96.60±0.18 | 73.46±0.56 |
| | NAGphormer | 71.51±0.13 | 77.42±1.41 | 89.70±0.19 | 91.22±0.14 | 95.49±0.11 | 95.75±0.09 | 97.34±0.03 | 77.16±0.72 |
| | Exphormer | 69.09±0.72 | 76.83±1.24 | 89.52±0.54 | 91.59±0.31 | 95.27±0.42 | 95.77±0.15 | 97.16±0.13 | 78.54±0.49 |
| | VCR-Graphormer | 71.67±0.10 | – | 89.77±0.15 | 91.75±0.15 | 95.53±0.14 | 95.37±0.04 | 97.34±0.04 | – |
| | GQT (ours) | **71.81±0.21** | **77.84±0.94** | **90.14±0.16** | **93.37±0.44** | **95.73±0.18** | **96.11±0.09** | **97.53±0.06** | **80.14±0.57** |

GNN and GT models on 7 out of 8 benchmarks (Table 2). Notably, this achievement comes with a significant memory reduction. For example, on the Physics dataset with 34,493 nodes, we only use $256 \times 6$ tokens, i.e., a 23-fold memory reduction.

**Heterophilic Node Classification.** We also evaluate GQT on six medium-scale heterophilic datasets: Squirrel, Chameleon (Rozemberczki et al., 2021), Questions, Roman-Empire, Amazon-Ratings, and Minesweeper (Platonov et al., 2023). We compare the performance with seven GNNs: GCN, GraphSAGE, GAT, GPRGNN, H2GCN (Zhu et al., 2020a), CPGNN (Zhu et al., 2021), and GloGNN (Li et al., 2022), and six GTs: GraphGPS, GOAT, NodeFormer, SGFormer, NAGphormer, and Exphormer. The baseline performance is reported from (Wu et al., 2023b; Luo et al., 2024b; Platonov et al., 2023; Behrouz & Hashemi, 2024). As shown in Table 3, GQT outperforms the baselines on five out of six datasets. We observe that introducing semantic edges and structural gating specifically benefits the heterophilic setting (Appendix E.1), as they enable the Transformer to capture long-range dependencies that are not easily accessible through the original graph structure.

**Large-scale Node Classification** We also use four large-scale datasets: ogbn-proteins, ogbn-arxiv, ogbn-products (Hu et al., 2020a), and pokec (heterogeneous) (Leskovec & Krevl, 2014). We compare the performance against six GNN: LINKX (Lim et al., 2021), SIGN (Frasca et al., 2020), GCN, GAT, GraphSAGE, and GPRGNN; and six GTs: GraphGPS, GOAT, NodeFormer, NAGphormer, Exphormer, and SGFormer (Wu et al., 2023b). We report the baseline performance from (Wu et al., 2023b; Luo et al., 2024a). The results (Table 4) show that GQT outperforms the baseline models on all large-scale benchmarks. This achievement comes with a significant reduction in required memory. For instance, on the ogbn-products dataset with 2,449,029 nodes and 100-dimensional node features, GQT requires only 3 codebooks of size 4096, resulting in a 30-fold memory reduction.

Table 3: Mean node classification performance on heterophilic graphs over five runs.

| | | Squirrel | Chameleon | Amazon-Ratings | Roman-Empire | Minesweeper | Questions |
|---|---|---|---|---|---|---|---|
| Dataset | #Nodes | 5,201 | 2,277 | 22,662 | 24,492 | 10,000 | 48,921 |
| | #Edges | 216,933 | 36,101 | 32,927 | 93,050 | 39,402 | 153,540 |
| | #Features | 2,089 | 2,325 | 300 | 300 | 7 | 301 |
| | #Classes | 5 | 5 | 18 | 5 | 2 | 2 |
| | Measure | Accuracy↑ | Accuracy↑ | Accuracy↑ | Accuracy↑ | ROC-AUC↑ | ROC-AUC↑ |
| GNN | GCN | 38.67±1.84 | 41.31±3.05 | 48.70±0.63 | 73.69±0.74 | 89.75±0.52 | 76.09±1.27 |
| | GraphSAGE | 36.09±1.99 | 37.77±4.14 | 53.63±0.39 | 85.74±0.67 | 93.51±0.57 | 76.44±0.62 |
| | GAT | 35.62±2.06 | 39.21±3.08 | 52.70±0.62 | 88.75±0.41 | 93.91±0.35 | 76.79±0.71 |
| | H2GCN | 35.10±1.15 | 26.75±3.64 | 36.47±0.23 | 60.11±0.52 | 89.71±0.31 | 63.59±1.46 |
| | CPGNN | 30.04±2.03 | 33.00±3.15 | 39.79±0.77 | 63.96±0.62 | 52.03±5.46 | 65.96±1.95 |
| | GPRGNN | 38.95±1.99 | 39.93±3.30 | 44.88±0.34 | 64.85±0.27 | 86.24±0.61 | 55.48±0.91 |
| | GloGNN | 35.11±1.24 | 25.90±3.58 | 36.89±0.14 | 59.63±0.69 | 51.08±1.23 | 65.74±1.19 |
| GT | GraphGPS | 39.67±2.84 | 40.79±4.03 | 53.10±0.42 | 82.00±0.61 | 90.63±0.67 | 71.73±1.47 |
| | NodeFormer | 38.52±1.57 | 34.73±4.14 | 43.86±0.35 | 64.49±0.73 | 86.71±0.88 | 74.27±1.46 |
| | SGFormer | 41.80±2.27 | **44.93±3.91** | 48.01±0.49 | 79.10±0.32 | 90.89±0.58 | 72.15±1.31 |
| | NAGphormer | 35.80±1.33 | – | 51.26±0.72 | 74.34±0.77 | 84.19±0.66 | – |
| | Exphormer | 36.04±1.45 | – | 53.51±0.46 | 89.03±0.37 | 90.74±0.53 | – |
| | GQT(ours) | **42.72±1.69** | 44.23±3.05 | **54.32±0.41** | **90.98±0.24** | **97.36±0.35** | **78.94±0.86** |

Table 4: Mean node classification performance on large-scale datasets over five runs.

| | | ogbn-proteins | ogbn-arxiv | ogbn-products | pokec |
|---|---|---|---|---|---|
| Dataset | #Nodes | 132,534 | 169,343 | 2,449,029 | 1,632,803 |
| | #Edges | 39,561,252 | 1,166,243 | 61,859,140 | 30,622,564 |
| | #Features | 128 | 8 | 100 | 65 |
| | #Classes | 40 | 2 | 47 | 2 |
| | Measure | ROC-AUC↑ | Accuracy ↑ | Accuracy ↑ | Accuracy ↑ |
| GNN | GCN | 72.51±0.35 | 71.74±0.29 | 75.64±0.21 | 75.45±0.17 |
| | GAT | 72.02±0.44 | 71.95±0.36 | 79.45±0.59 | 72.23±0.18 |
| | GPRGNN | 75.68±0.49 | 71.10±0.12 | 79.76±0.59 | 72.23±0.18 |
| | LINKX | 71.37±0.58 | 66.18±0.33 | 71.59±0.71 | 82.04±0.07 |
| | GraphSAGE | 77.68±0.20 | 71.49±0.27 | 78.29±0.16 | 75.63±0.38 |
| | SIGN | – | 71.95±0.11 | 80.52±0.16 | – |
| GT | GraphGPS | 76.83±0.26 | 70.97±0.41 | OOM | OOM |
| | GOAT | 74.18±0.37 | 72.41±0.40 | 82.00±0.43 | 66.37±0.94 |
| | NodeFormer | 77.45±1.15 | 59.90±0.42 | 72.93±0.13 | 71.00±1.30 |
| | SGFormer | 79.53±0.38 | 72.63±0.13 | 74.16±0.31 | 73.76±0.24 |
| | NAGphormer | 73.61±0.33 | 70.13±0.55 | 73.55±0.21 | 76.59±0.25 |
| | Exphormer | 74.58±0.26 | 72.44±0.28 | OOM | OOM |
| | GQT(ours) | **82.13±0.34** | **73.14±0.16** | **82.46±0.17** | **83.76±0.24** |

## 6.2 ABLATION STUDY

**Effect of Tokenization**. We examine the performance of the tokenizer by training a linear model on the representations of the learned tokens without modulation, augmentation, or Transformer (1). As shown in Table 6, within the linear evaluation protocol, the tokenizer shows strong performance, surpassing that of GTs such as GraphGPS and NAGphormer, as well as GNNs like GAT and SIGN (Table 4). This implies that the tokenizer is capable of learning effective token representations. To further investigate the importance of the tokenizer, we exclude it and train the Transformer directly on the original node features (2). As expected, this results in significant degradation in performance, highlighting the crucial role of the tokenizer. Additionally, to study the effects of vector quantization, GraphMAE2, and DGI objectives, we train the model by excluding each component (3-5). The results suggest that the SSL objectives contribute more significantly to the performance compared to vector quantization. This is because the primary purpose of vector quantization is to compress information into discrete tokens, reducing memory requirements. Between GraphMAE2 and DGI, the former

Table 6: Ablation study on effect of proposed components on the ogbn-arxiv dataset.

| | Graph Tokenizer | | | Token Modulation | | | Augmentation | | Model | Performance |
|---|---|---|---|---|---|---|---|---|---|---|
| | RVQ | GMAE2 | DGI | Codebook Embeddings | Positional Encoding | Structural Gating | Semantic Edges | PPR Sequence | | Accuracy↑ |
| (1) | ✓ | ✓ | ✓ | ✓ | | | | | Linear | 71.91±0.13 |
| (2) | | | | | ✓ | | | ✓ | Transformer | 70.68±0.17 |
| (3) | | ✓ | ✓ | ✓ | ✓ | ✓ | ✓ | ✓ | Transformer | 72.84±0.23 |
| (4) | ✓ | | ✓ | ✓ | ✓ | ✓ | ✓ | ✓ | Transformer | 71.83±0.19 |
| (5) | ✓ | ✓ | | ✓ | ✓ | ✓ | ✓ | ✓ | Transformer | 72.71±0.24 |
| (6) | ✓ | ✓ | ✓ | | ✓ | ✓ | ✓ | ✓ | Transformer | 71.34±0.16 |
| (7) | ✓ | ✓ | ✓ | ✓ | | ✓ | ✓ | ✓ | Transformer | 72.69±0.21 |
| (8) | ✓ | ✓ | ✓ | ✓ | ✓ | | ✓ | ✓ | Transformer | 73.08±0.14 |
| (9) | ✓ | ✓ | ✓ | ✓ | ✓ | ✓ | | ✓ | Transformer | 72.59±0.25 |
| (10) | ✓ | ✓ | ✓ | ✓ | ✓ | ✓ | ✓ | ✓ | Transformer | **73.14±0.16** |

yields the highest gain. This is due to its composition of two objectives: masked reconstruction and teacher-(noisy)student distillation. Both of these objectives have been shown to outperform InfoMax objectives on downstream tasks (Hou et al., 2022; Thakoor et al., 2022).

**Effect of Modulation**. We also investigate the impact of codebook embeddings, positional encoding, and structural gating on the model's performance (6-8). As shown in Table 6, introducing aggregated codebook embeddings leads to improved downstream performance because it provides the Transformer with richer representations of each token. Positional encoding, as observed in other domains, contributes moderately to downstream performance. We also note that introducing structural gating yields moderate improvements in homophilic settings, whereas the gains are significant in heterophilic benchmarks (E.1). This disparity can be attributed to the ability of structural gating to provide the Transformer with importance scores computed over the global graph structure, which is particularly beneficial in heterophilic scenarios.

**Effect of Augmentation**. We study the effect of semantic edges on downstream performance (9). The results suggest that augmenting the graph structure with semantic edges yields significant gains. This is because introducing semantic edges allows the Transformer to access semantic information that may not be captured by the original graph structure. Furthermore, when combined with random walks, this also enables the Transformer to attend to long-range dependencies which is particularly important in heterophilic benchmarks, where semantic relationships between nodes are more nuanced.

**Robustness Analysis**. To measure robustness, we use Greedy Randomized Block Coordinate Descent (GRBCD) and Projected Randomized Block Coordinate Descent (PR-BCD) (Geisler et al., 2021) adversarial attacks to measure the accuracy degradation. We compare the GQT with an RQ-VAE (Lee et al., 2022). The results (Table 5) suggest that our tokenizer is more robust to attacks. This is because GQT is trained with multi-task self-supervised objectives while RQ-VAE is trained with autoencoding objective. More details are provided in Appendix E.

Table 5: Accuracy drop under adversarial attack.

| Attack | GR-BCD | | PR-BCD | |
|---|---|---|---|---|
| | PubMed | ogbn-arxiv | PubMed | ogbn-arxiv |
| RQ-VAE | 20.40% | 14.80% | 23.30% | 17.20% |
| GQT (ours) | 15.80% | 10.40% | 18.10% | 11.30% |

## 7 CONCLUSION

We introduced the **G**raph **Q**uantized **T**okenizer (GQT) to provide standard Transformer encoders to acces discrete graph tokens that encapsulate local interactions and allow Transformers to attend to long-range dependencies within the graph structure. This allows us to seamlessly take advantage of the rapid advances in scaling Transformers. We achieved state-of-the-art performance on 20 out of 22 datasets, including large-scale and long-range homophilic and heterophilic datasets. As future directions, we plan to explore the potential of GQT in generative graph learning. Additionally, we aim to couple GQT with LLMs to provide a shared feature space across various graph datasets, paving the way for true Graph Foundational Models (GFMs) (Liu et al., 2023b; Mao et al., 2024).

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

# Appendix

## A  PRELIMINARIES

**Graph Attention Networks (GAT)**. The representation of node $i$ in layer $l$ is computed as:

$$h_i^l = \sigma \left( \sum_{j \in \mathcal{N}_i} \alpha_{ij} \mathbf{W} h_j^{(l-1)} \right), \quad \alpha_{ij} = \frac{\exp \left( \sigma \left( \mathbf{W}_2 \left[ \mathbf{W}_1 h_i^{(l-1)} \| \mathbf{W}_1 h_j^{(l-1)} \right] \right) \right)}{\sum_{k \in \mathcal{N}_i} \exp \left( \sigma \left( \mathbf{W}_2 \left[ \mathbf{W}_1 h_i^{(l-1)} \| \mathbf{W}_1 h_k^{(l-1)} \right] \right) \right)} \quad (17)$$

where $\sigma$ is a non-linearity, and $\alpha_{ij}$ is the normalized attention score between two connected nodes $i$ and $j$.

**Personalized PageRank (PPR)**. A PPR vector for a node $u$ captures the relative importance of other nodes with respect to node $u$ by exploring the graph structure through iterative random walks:

$$r = \alpha \mathbf{P} r + (1 - \alpha) q \quad (18)$$

where $\mathbf{P} = \mathbf{D}^{-\frac{1}{2}} \mathbf{A} \mathbf{D}^{-\frac{1}{2}} \in \mathbb{R}^{n \times n}$, $q$ is a stochastic personalized vector, $r$ is the stationary distribution of random walks, and $\alpha$ is a damping factor.

## B  MODEL DETAILS

---

**Algorithm 1** Graph Tokenizer

---

1: **Input:** Graph $g = (\mathcal{V}, \mathcal{E}, \mathbf{X})$, Graph Encoder $\text{GNN}_\theta$, Residual Quantizer $\text{RQ}_\Phi$, BGRL Loss $\text{RQ}_\Phi$

2: $\mathbf{H_v} = \text{GNN}_\theta(g)$                    {Node Representations}
3: $\mathbf{C}, \mathbf{Z}, \mathbf{T}, \mathcal{L}_{commit} = \text{RQ}_\Phi(\mathbf{H_v})$    {codebooks, quantized representation, discrete tokens}
4: $\mathcal{L}_{dgi} = \text{DGI}(\mathbf{Z})$                    {Compute DGI Loss}
5: $\mathcal{L}_{bgrl} = \text{BGRL}(\mathbf{Z})$                    {Compute BGRL Loss}
6: $\mathcal{L}_{mae} = \text{MAE}(\mathbf{Z})$                    {Compute MAE Loss}
7: $\mathcal{L} = \mathcal{L}_{dgi} + \mathcal{L}_{bgrl} + \mathcal{L}_{mae} + \beta \times \mathcal{L}_{commit}$    {Compute Multi-Task Loss}
8: **return** $\mathbf{Z}, \mathbf{T}, \mathcal{L}$

---

**Algorithm 2** Residual Vector Quantization

---

1: **Input:** Data $\mathbf{X}$, Number of codebooks $N$, Size of codebook $K$, Dimension of codebooks $d$
2: $\mathcal{L}_{commit} = 0$
3: $\mathbf{C} = \text{Random}(N, K, d)$
4: $\mathbf{Z} = \text{Zeros}(|\mathbf{X}|, d)$
5: $\mathbf{T} = \text{Zeros}(|\mathbf{X}|, N)$
6: **for** $i$ in $|\mathbf{X}|$ **do**
7:   $r = \mathbf{X}[i]$
8:   **for** $j = 1$ to $N$ **do**
9:     $k = \arg\min_k \|r - \mathbf{C}[j, k]\|_2^2$
10:     $r = \|r - \mathbf{C}[j, k]\|_2^2$
11:     $\mathbf{T}[i][j] = k$
12:     $\mathbf{Z}[i] = \mathbf{Z}[i] + \mathbf{C}[j][k]$
13:     $\mathcal{L}_{commit} = \mathcal{L}_{commit} + \|r - \text{sg}[\mathbf{C}[j, k]]\|_2^2$
14:   **end for**
15: **end for**
16: **return** $\mathbf{Z}, \mathbf{T}, \mathcal{L}/|\mathbf{X}|$

---

## C  DATASETS

We provide a detailed description of the datasets used in this study. All datasets are publicly available.

- **CoraFull** (Bojchevski & Günnemann, 2017), **CiteSeer**, and **Pubmed** (Namata et al., 2012) are citation datasets, where nodes represent documents and edges represent citation links. Labels indicate the paper category.

- **Computer** and **Photo** (Shchur et al., 2018) are from the Amazon co-purchase graph (McAuley et al., 2015), where nodes represent goods and edges indicate that two goods are frequently bought together. Node features are bag-of-words encoded product reviews, and class labels are given by the product category.

- **CS** and **Physics** (Shchur et al., 2018) are co-authorship graphs based on the Microsoft Academic Graph from the KDD Cup 2016 challenges. Here, nodes are authors connected by an edge if they co-authored a paper; node features represent paper keywords for each author's papers, and class labels indicate the most active fields of study for each author.

- **WikiCS** (Mernyei & Cangea, 2020) is derived from Wikipedia, where nodes represent Computer Science articles, and edges are based on hyperlinks. The nodes are classified into 10 classes representing different branches of the field.

- **Squirrel** and **Chameleon** (Rozemberczki et al., 2021; Pei et al., 2020) are Wikipedia page-page networks, where nodes represent articles from the English Wikipedia, and edges reflect mutual links between them. The nodes were classified into five classes based on their average monthly traffic.

- **Amazon-Ratings** (Platonov et al., 2023) is based on Amazon product co-purchasing data. Nodes represent products (books, music CDs, DVDs, VHS video tapes), and edges connect products that are frequently bought together. The task is to predict the average rating given to a product by reviewers.

- **Roman-Empire** (Platonov et al., 2023) is based on the Roman Empire article from the English Wikipedia. Each node in the graph corresponds to one word (not necessarily unique) in the text, so the number of nodes equals the length of the article. Two words are connected if they follow each other in the text or are linked in the sentence's dependency tree. A node's class represents its syntactic role.

- **Minesweeper** (Platonov et al., 2023) is inspired by the Minesweeper game. The graph consists of regular 100x100 grid, where each node (cell) is connected to eight neighboring nodes (except for nodes at the edge of the grid, which have fewer neighbors). 20% of the nodes are randomly selected as mines. The task is to predict which nodes are mines. Node features are one-hot-encoded numbers of neighboring mines, however, for 50% of the nodes, these features are unknown, indicated by a separate binary feature.

- **Questions** (Platonov et al., 2023) is based on data from the question-answering website Yandex Q, where nodes represent users, and edges connect two nodes if one user answered another user's question during a one-year time interval. The task is to predict which users remained active on the website, forming a binary classification task.

- **ogbn-proteins** (Hu et al., 2020a) is a protein-protein association network, where nodes represent proteins, and edges indicate biologically meaningful associations between proteins, such as physical interactions, co-expression, or homology. The task is to predict the presence of protein functions in a multi-label binary classification setup.

- **ogbn-arxiv** (Hu et al., 2020a) is a citation network between all Computer Science (CS) arXiv papers indexed by MAG (Wang et al., 2020). Each node presents an arXiv paper, and directed edges indicate that one paper cites another. The task is to predict the 40 subject areas of arXiv CS papers, such as cs.AI, cs.LG, and cs.OS.

- **ogbn-products** (Hu et al., 2020a) is an Amazon product co-purchasing network[1] of 2 million products. Edges indicate that products are purchased together. The task is to predict the product category.

- **pokec** (Leskovec & Krevl, 2014; Lim et al., 2021) is a social network, where nodes represent users, and edges represent friendships. The task is to predict the gender of users.

- **Peptides-Func** is a peptide dataset retrieved from SATPdb (Singh et al., 2016) with over 15K peptides. Each node corresponds to a heavy atom, and edges are chemical bonds. The task is to predict 10 peptide functions, forming a multi-label graph classification task.

---

[1] http://manikvarma.org/downloads/XC/XMLRepository.html

- **Peptides-Struct** consists the same graphs as Peptides-Struct, but with different task. Here the task is to predict aggregated 3D properties (i.e., mass, valence) of the peptides at the graph level.

- **COCO-SP** is a node classification dataset based on the MC COCO image dataset (Lin et al., 2014). Each node corresponds to a region of the image belonging to a particular class. These superpixels nodes are extracted with the SLIC algorithm (Achanta et al., 2012), and two nodes are connected with an edge if the node regions share a common boundary. The task is to predict the semantic segmentation label for each superpixel node out of 81 classes.

- **PCQM-Contact** is a molecule dataset with over 529K molecules (Nakata & Shimazaki, 2017). Atoms are nodes, and chemical bonds are edges. The task is to predict pairs of nodes that will be contacting with each other in the 3D space.

For CoraFull, Pubmed, PubMed, Computer, Photo, CS, and Physics, we follow previous work and use 60%/20%/20% train/valid/test split. For WiKiCS, we follow the official split in Mernyei & Cangea (2020). For Squirrel, Chameleon, Amazon-Ratings, Roman-Empire, Minesweeper, and Questions, we follow the splits in Platonov et al. (2023). For ogbn-proteins, ogbn-arxiv, and ogbn-products, we follow the splits in Hu et al. (2020a). For pokec, we follow the split used in Lim et al. (2021). For Peptides-Func, Peptides-Struct, COCO-SP, and PCQM-Contact, we follow the split provided in Dwivedi et al. (2022b).

## D   EXPERIMENTAL SETUP

**Software & Hardware.** GQT is implemented using PyTorch[2], PyG[3], DGL[4], and the vector-quantize-pytorch package[5]. Most datasets can be accessed through PyG and DGL. All experiments are conducted on a single Nvidia A100 GPU.

**Hyperparameters & Experimental Details.** As illustrated in Figure 1, our method consists of two parts: the tokenizer and the Transformer encoder. We provide the hyperparameters and experimental details for each part below.

During the training of the graph tokenizer, we use full-graph training for small- and medium-scale datasets, and apply sampling for large-scale graphs. We consider different sampling methods, including random partitioning, which randomly samples nodes within a graph and returns their induced subgraph; neighbor sampling (Hamilton et al., 2017), GraphSAINT (Zeng et al., 2020), and local clustering Hou et al. (2023). For the GNN encoder and decoder, we use GCN or GAT as our backbone and tune the number of layers from {1, 2, 3, 4, 5, 6, 7, 8, 9, 10} and hidden dimensions from {128, 256, 512, 1024}. For the quantizer, we use residual-VQ (RVQ) (Lee et al., 2022) and tune the number of codebooks from {1, 2, 3, 6, 9} and the codebook size from {128, 256, 512, 1024, 2048, 4096}. We set the code dimension to be equal to the hidden dimension of the GNN encoder.

During the training of the Transformer, we use KNN to add semantic edges and tune the number of semantic neighbors from {0, 5, 10, 15, 20}. Then, we use PPR to generate a sequence of nodes for each target node. We tune the number of PPR neighbors from {0, 5, 10, 20, 30, 50}. For the Transformer model, we use the TransformerEncoder module in PyTorch as our backbone, and tune the number of layers from{1, 2, 3, 4, 5, 6}, the number of heads from {4, 8}, and the feedforward dimension from {512, 1024, 2048}. Note that for some small- and medium-scale datasets, we do not need PPR to generate sequences, instead, we can directly take all nodes from one graph as a sequence as in Rampášek et al. (2022).

---

[2]https://pytorch.org/
[3]https://pyg.org/
[4]https://www.dgl.ai/
[5]https://github.com/lucidrains/vector-quantize-pytorch

Table 7: Selected hyperparameters for each dataset.

| | **GNN Encoder** | | **Quantizer** | | | | **Transformer** | | |
|---|---|---|---|---|---|---|---|---|---|
| | # layers | # Hidden dim | # Codebooks | Codebook size | KNN | PPR | # Layers | # Heads | # FFN dim |
| CoraFull | 2 | 256 | 3 | 128 | 0 | 15 | 2 | 4 | 512 |
| CiteSeer | 2 | 256 | 3 | 128 | 5 | 15 | 2 | 4 | 512 |
| PubMed | 2 | 256 | 3 | 256 | 0 | 15 | 2 | 4 | 512 |
| Computer | 2 | 256 | 3 | 128 | 5 | 30 | 2 | 4 | 512 |
| Photo | 3 | 512 | 3 | 128 | 5 | 30 | 2 | 4 | 1024 |
| CS | 2 | 512 | 3 | 128 | 5 | 20 | 2 | 4 | 1024 |
| Physics | 2 | 256 | 3 | 256 | 5 | 30 | 2 | 4 | 512 |
| WikiCS | 2 | 256 | 3 | 128 | 5 | 30 | 2 | 4 | 512 |
| Squirrel | 3 | 256 | 3 | 128 | 5 | 30 | 2 | 4 | 512 |
| Chameleon | 3 | 256 | 3 | 128 | 5 | 30 | 2 | 4 | 512 |
| Amazon-Ratings | 4 | 512 | 3 | 128 | 5 | 20 | 2 | 4 | 1024 |
| Roman-Empire | 6 | 256 | 3 | 256 | 10 | 15 | 3 | 4 | 512 |
| Minesweeper | 6 | 128 | 3 | 128 | 10 | 15 | 2 | 4 | 512 |
| Questions | 3 | 256 | 3 | 512 | 10 | 15 | 2 | 4 | 512 |
| ogbn-proteins | 6 | 256 | 3 | 512 | 0 | 50 | 3 | 4 | 512 |
| ogbn-arxiv | 4 | 512 | 3 | 512 | 5 | 30 | 2 | 4 | 1024 |
| ogbn-products | 4 | 1024 | 3 | 4096 | 5 | 30 | 2 | 8 | 2048 |
| pokec | 6 | 256 | 3 | 512 | 0 | 50 | 3 | 4 | 512 |
| Peptides-Func | 4 | 128 | 3 | 128 | 0 | 0 | 2 | 4 | 512 |
| Peptides-Struct | 4 | 128 | 3 | 128 | 0 | 0 | 2 | 4 | 512 |
| COCO-SP | 4 | 128 | 3 | 128 | 0 | 0 | 2 | 4 | 512 |
| PCQM-Contact | 4 | 128 | 3 | 128 | 0 | 0 | 2 | 4 | 512 |

# E    ADDITIONAL RESULTS

## E.1    FURTHER ABLATION STUDY

We also provide an ablation study on one of the heterophilic datasets. The results shown in Table 8 suggest that introducing semantic edges and structural gating mechanisms specifically benefits the heterophilic setting.

Table 8: Ablation study on effect of proposed components on the Minesweeper dataset.

| | **Graph Tokenizer** | | | **Token Modulation** | | | **Augmentation** | | **Model** | **Performance** |
|---|---|---|---|---|---|---|---|---|---|---|
| | RVQ | GMAE2 | DGI | Codebook Embeddings | Positional Encoding | Structural Gating | Semantic Edges | PPR Sequence | | ROC-AUC↑ |
| (1) | ✓ | ✓ | ✓ | ✓ | | | | | Linear | 90.24±0.49 |
| (2) | | | | | ✓ | | | ✓ | Transformer | 90.52±0.39 |
| (3) | | ✓ | ✓ | ✓ | ✓ | ✓ | ✓ | ✓ | Transformer | 95.27±0.46 |
| (4) | ✓ | | ✓ | ✓ | ✓ | ✓ | ✓ | ✓ | Transformer | 92.91±0.55 |
| (5) | ✓ | ✓ | | ✓ | ✓ | ✓ | ✓ | ✓ | Transformer | 93.82±0.46 |
| (6) | ✓ | ✓ | ✓ | | ✓ | ✓ | ✓ | ✓ | Transformer | 93.24±0.36 |
| (7) | ✓ | ✓ | ✓ | ✓ | | ✓ | ✓ | ✓ | Transformer | 94.82±0.41 |
| (8) | ✓ | ✓ | ✓ | ✓ | ✓ | | ✓ | ✓ | Transformer | 93.97±0.58 |
| (9) | ✓ | ✓ | ✓ | ✓ | ✓ | ✓ | | ✓ | Transformer | 92.83±0.35 |
| (10) | ✓ | ✓ | ✓ | ✓ | ✓ | ✓ | ✓ | ✓ | Transformer | 95.28±0.44 |

## E.2    GENERALIZATION ANALYSIS

To measure improved generalization, we follow the common practice of treating downstream predictive performance as a proxy for generalization. As shown in Table 6 and Table 8, every component of the tokenizer, including both SSL objectives and the quantization layer, contributes to the downstream predictive performance, thereby improving the model's generalizability. Furthermore, to evaluate the contribution of multi-task SSL objectives to downstream performance, we compare our results with

Table 9: Comparison between mean GQT and RQ-VAE performance over five runs.

| | ogbn-arxiv | Minesweeper |
|---|---|---|
| RQ-VAE | 66.05±0.48 | 89.69±0.35 |
| GQT (ours) | 73.14±0.16 | 95.28±0.44 |

those of a tokenizer trained using the RQ-VAE (Lee et al., 2022) design, which employs a reconstruction objective. The results presented below indicate that using multi-task SSL objectives significantly improves downstream predictive performance, which is strongly correlated with the method's generalization.

## E.3 Efficiency Analysis

As mentioned in Section 6.1, using discrete tokens instead of node features results in significant memory reduction. For instance, on the ogbn-products dataset with 2,449,029 nodes and 100-dimensional node features, GQT requires only 3 codebooks of size 4096, resulting in a remarkable 30-fold reduction in memory usage.

Table 10: Memory and run time during inference.

| Attack | GPU Memory | | Full Inference Time | |
|---|---|---|---|---|
| | ogbn-arxiv | Minesweeper | ogbn-arxiv | Minesweeper |
| GAT | 2715MB | 2108M | 5s | 1s |
| GQT (ours) | 1324MB | 1037MB | 4s | 1s |

This memory reduction occurs after training the tokenizer. Since the encoder of the tokenizer is a GNN that processes the graph with original node features, its memory footprint is comparable to that of any arbitrary GNN. However, because the Transformer encoder only consumes discrete tokens, which are significantly fewer than the total number of nodes, we achieve a substantial reduction in memory footprint. As an additional experiment, we compare the inference time and memory usage between our Transformer encoder and a Graph Attention Network (GAT) when performing inference on all graph nodes. The results shown in Table 10 show that while our Transformer is on par with a sparse implementation of GAT in terms of inference time, it requires half the GPU memory.

