# LEARNING GRAPH QUANTIZED TOKENIZERS FOR TRANSFORMERS

## ABSTRACT

Transformers serve as the backbone architectures of Foundational Models, where a domain-specific tokenizer helps them adapt to various domains. Graph Transformers (GTs) have recently emerged as a leading model in geometric deep learning, outperforming Graph Neural Networks (GNNs) in various graph learning tasks. However, the development of tokenizers for graphs has lagged behind other modalities, with existing approaches relying on heuristics or GNNs co-trained with Transformers. To address this, we introduce GQT (**G**raph **Q**uantized **T**okenizer), which decouples tokenizer training from Transformer training by leveraging multi-task graph self-supervised learning, yielding robust and generalizable graph tokens. Furthermore, the GQT utilizes Residual Vector Quantization (RVQ) to learn hierarchical discrete tokens, resulting in significantly reduced memory requirements and improved generalization capabilities. By combining the GQT with token modulation, a Transformer encoder achieves state-of-the-art performance on 16 out of 18 benchmarks, including large-scale homophilic and heterophilic datasets.

## 1 INTRODUCTION

Following the success of Transformers (Vaswani et al., 2017) in natural language processing (Devlin et al., 2019; Brown et al., 2020) and computer vision (Dosovitskiy et al., 2021), Graph Transformers (GTs) (Dwivedi & Bresson, 2020; Ying et al., 2021a; Rampášek et al., 2022; Ma et al., 2023; Shirzad et al., 2023; Kong et al., 2023b; Chen et al., 2023; Wu et al., 2022b) have emerged as strong models in geometric deep learning. Unlike message-passing Graph Neural Networks (GNNs), which rely on strong locality inductive biases (Battaglia et al., 2018; Veličković et al., 2018; Hou et al., 2020; Hamilton et al., 2017a; Kipf & Welling, 2017), GTs are inherently more expressive due to their ability to capture long-range interactions between nodes (Ma et al., 2023). This is particularly beneficial in heterophilous settings where local alignment does not hold (Fu et al., 2024). GTs possess an expressive power at least equivalent to the 2-Weisfeiler-Lehman (WL) isomorphism test (Kim et al., 2022), which is sufficient for most real-world tasks (Zopf, 2022). This surpasses the expressive power of message-passing GNNs, which are limited to the 1-WL test (Ying et al., 2021a). Furthermore, a Transformer with sufficient attention heads can match or exceed the expressive power of a second-order invariant graph network, outperforming message-passing GNNs (Kim et al., 2022). However, both GNNs and Transformers are susceptible to over-smoothing (Li et al., 2018; Zhou et al., 2021; Dovonon et al., 2024).

GTs require consideration of both graph structure and features, as nodes with identical features will otherwise be projected into the same representation regardless of their surrounding structures (Hoang et al., 2024). There are three general approaches to address this limitation (Hoang et al., 2024): (1) node feature modulation, which involves injecting structural information into the node features; (2) context node sampling, where a sampling strategy is used to construct a sequence over the neighbor nodes; and (3) modifying the architecture of a vanilla Transformer to directly incorporate structural biases. Given that Transformers are universal approximators of sequence-to-sequence functions (Yun et al., 2020) and considering the rapid developments in efficient implementation of multi-head attention (MHA) module (Dao et al., 2022a; Liu et al., 2024), which enables longer context sizes of up to million-scale tokens (Reid et al., 2024), a well-designed graph tokenizer can allow a vanilla Transformer model to efficiently process even large-scale graphs. Recent studies on applying Large Language Models (LLMs) to graph-related tasks have found that representing graphs through textual descriptions can lead to surprisingly strong performance gains that surpass those of GNNs, suggesting

that vanilla Transformers are indeed capable of effectively learning graph structures (Ye et al., 2024; He et al., 2024). Nonetheless, LLMs are not inference-efficient, and hence our goal in this paper

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

$$h_v^l = \text{LN} \left( \text{MHA} \left( \text{LN} \left( h_v^{l-1} \right) \right) + h_v^{l-1} \right) \tag{2}$$

$$h_v^l = h_v^l + \text{MLP} \left( h_v^l \right) \tag{3}$$

where LN and MHA are Layer Normalization and Multi-Head Attention modules, respectively. Similar to Transformer encoders in other modalities (Devlin et al., 2019; Dosovitskiy et al., 2021), we can append a special classification token, denoted as [CLS], to the input and use its representation to perform various classification tasks on the graph: MLP $(h_{\text{[CLS]}})$. In this setting, the input for node classification is $T_v$, for link prediction is $[T_v || T_u]$, and for graph classification is $[T_v ||_{v \in \mathcal{V}}]$.

**Vector Quantization** projects embeddings $\mathbf{X} \in \mathbb{R}^{n \times d_x}$ into a more compact space of codebooks $\mathbf{C} \in \mathbb{R}^{k \times d_c}$, where $k \ll n$. The codebooks can be learned by minimizing various objectives such as K-means clustering. The new representation of $x_i$ is then computed as follows (Van Den Oord et al., 2017):

$$z(x_i) = c_k \quad \text{where} \quad k = \arg\min_j \|x_i - c_j\|_2^2 \tag{4}$$

Building upon this concept, RQ-VAE (Lee et al., 2022) extends VQ to a sequence of codebooks, where each consecutive codebook quantizes the residual error from the previous codebook, i.e.,

$r_i = z_i - c_k$. This hierarchical approach constructs a multi-level quantized representation, enhancing the overall quantization quality.

# 4 SELF-SUPERVISED GRAPH TOKENIZATION

## 4.1 TOKENIZER PROPERTIES

Our goal is to design a graph tokenizer that can learn to generate tokens that exhibit three key characteristics, which are essential for effective graph representation learning. These characteristics are as follows.

**Local Interactions**. The learned tokens should encapsulate local interactions, allowing the Transformer to focus on global dependencies. This is analogous to Vision Transformers (ViTs), where the Transformer attends to image patches instead of pixels, enabling efficient learning on abstract tokens (Dosovitskiy et al., 2021; Liu et al., 2021). To achieve a similar effect on graph-structured data, we leverage message-passing GNNs as the foundation of the tokenizer's encoder, capitalizing on their strong locality inductive bias to effectively capture local interactions in the representation space (Battaglia et al., 2018). Our design accommodates various GNN layer choices without constraints; for simplicity, we opt for the widely used Graph Attention Network (GAT) (Veličković et al., 2018) as our base graph encoder. The representation of node $i$ in layer $l$ is computed as:

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

## 4.2 TRAINING

The GNN encoder is trained through gradient descent to minimize a loss function comprising three terms, where $\beta$ is the loss weight:

$$\mathcal{L} = \mathcal{L}_{\text{dgi}} + \mathcal{L}_{\text{gmae2}} + \beta \mathcal{L}_{\text{commit}} \tag{6}$$

The first term is the DGI objective, which maximizes mutual information (MI) between node representations and graph (sub-graph) representations, based on the Jensen-Shannon divergence between the joint and product of marginals as follows (Veličković et al., 2019):

$$\mathcal{L}_{\text{dgi}} = \mathbb{E}\left(\sum_{v \in g} \log\left(\mathcal{D}\left(h_v, h_g\right)\right) + \sum_{u \in \tilde{g}} \log\left(1 - \mathcal{D}\left(\tilde{h}_u, h_g\right)\right)\right) \tag{7}$$

where $h_u$ is the representation of node $u$, $h_g$ is the patch (graph/sub-graph) representation that the node belongs to, $\mathcal{D}(.,.)$ is a discriminator computing the probability scores between local and global information, and $\tilde{g}$ is the corrupted version of the original graph providing negative examples. Following (Veličković et al., 2019), we define the discriminator as a bilinear layer $\mathcal{D}\left(h_u, h_g\right) = \sigma\left(h_u^T \mathbf{W} h_g\right)$, compute the global representation as a mean of node representations: $h_g = \frac{1}{|\mathcal{V}|}\sum_{v \in g} h_v$, and define $\tilde{g}$ as a graph with the same structure but randomly shuffled features.

The second term is the GraphMAE2 objective (Hou et al., 2023), which combines the generative loss of GraphMAE (Hou et al., 2022) with the teacher-(noisy)student distillation loss of BGRL (Thakoor et al., 2022). This combination enables the model to avoid overfitting and learn more semantic representations. The GraphMAE2 loss is computed as follows:

$$\mathcal{L}_{\text{gmae2}} = \sum_{v \in \tilde{g}} \left(1 - \frac{x_v^T . \tilde{h}_v}{\|x_v^T\| . \|\tilde{h}_v\|}\right)^\gamma + \lambda \sum_{v \in g} \left(1 - \frac{h_v^T . \tilde{h}_v}{\|h_v^T\| . \|\tilde{h}_v\|}\right)^\gamma \tag{8}$$

where $\tilde{g}$ is the masked graph, $\tilde{h}_v$ is the node representation of a masked node learned by the noisy student, $h_v$ is the corresponding node representation learned by the teacher over the original graph, and $\gamma \geq 1$ is a scaling factor. Note that the teacher's parameters are updated using an exponential moving average (EMA) of the noisy student's parameters.

The third term is the commitment loss, which encourages the representations to get close to their corresponding codebook embeddings within the RVQ layer. This loss is computed as:

$$\mathcal{L}_{\text{commit}} = \frac{1}{|\mathcal{V}|} \sum_{v \in g} ||h_v - \text{sg}[c_k]||_2 \tag{9}$$

where sg is the stop-gradient operator, and $c_k$ is the representation of the codebook that $h_v$ is assigned to (i.e., the centroid or prototype vector). Note that this loss only affects the node representations and does not update the codebooks.

To initialize and update the codebooks, we employ K-Means clustering and EMA with weight decay $\tau \in [0, 1]$, respectively. Specifically, the codebooks are updated as follows:

$$c_k^t = \tau c_k^{t-1} + (1 - \tau)\frac{1}{|\mathcal{V}_k|} \sum_{v \in \mathcal{V}_k} h_v \tag{10}$$

where $\mathcal{V}_k$ is the set of nodes assigned to codebook $c_k$. This update rule allows the codebooks to adapt to the changing node representations while maintaining stability.

## 5 GRAPH TRANSFORMER

### 5.1 SEQUENCE GENERATION

Once the tokenizer is trained, each node $v \in \mathcal{V}$ is mapped to a set of $c$ tokens: $T_v = [t_1^v, \cdots, t_c^v] \in \mathbb{N}^c$, which compress information about local interactions. To enable the Transformer to capture long-range interactions, the input should consist of a sequence of tokens from nodes that are likely to have long-range dependencies. To facilitate this, we first augment the graph with *semantic edges* denoted as $\mathcal{E}_s$, which are computed as follows:

$$\mathcal{E}_s = \left\{ e_{u,v} \mid \underset{u \in \mathcal{V}}{\arg \text{topk}} \, \text{sim}\left(f\left(x_u\right), f\left(x_v\right)\right) \forall v \in \mathcal{V} \right\} \tag{11}$$

where $\text{sim}(\cdot, \cdot)$ denotes the similarity function, $x_u$ is the feature vector of node $u$, and $f$ is a projection function. We use cosine similarity as the similarity function and principal component analysis (PCA) as the projection function. This semantic edge augmentation effectively creates sparse edges between each node and its k-nearest neighbors in the feature space, enhancing the model's ability to recognize and utilize significant long-range dependencies.

We then merge the semantic edges with the original graph edges and use Personalized PageRank (PPR) to generate a sequence per node. A PPR vector for a node $u$ captures the relative importance of other nodes with respect to node $u$ by exploring the graph structure through iterative random walks:

$$r = \alpha \mathbf{P} r + (1 - \alpha)q \tag{12}$$

where $\mathbf{P} = \mathbf{D}^{-\frac{1}{2}}\mathbf{A}\mathbf{D}^{-\frac{1}{2}} \in \mathbb{R}^{n \times n}$, $q$ is a stochastic personalized vector, $r$ is the stationary distribution of random walks, and $\alpha$ is a damping factor.

Using PPR enriches the sequence with information beyond local interactions, allowing the Transformer to access potential long-range dependencies. We construct the sequence $S_v$ for each node $v$ as follows:

$$S_v = \left[T_v \| T_u \|_{u \in \arg \text{topk} \, \text{PPR}(v, \mathcal{E} \cup \mathcal{E}_s)}\right] \tag{13}$$

where $S_v = [t_1^v \cdots t_c^v \mid t_1^{u_1} \cdots t_c^{u_1} \mid \cdots \mid t_1^{u_k} \cdots t_c^{u_k}]$ is the sequence of sorted integer tokens with length $c \times (k + 1)$, based on the PPR scores for node $u$. Note that the computation of semantic edges and PPR sequences is performed only once as a pre-processing step, which reduces computational overhead during training.

## 5.2 TOKEN MODULATION

There are $c \times K$ possible integer tokens in total, where $c$ is the number of codebooks and $K$ is the codebook size. We randomly initialize an embedding matrix $\mathbf{X}_T \in \mathbb{R}^{c \times K \times d_x}$, which is trained end-to-end with the Transformer. To further enrich the token representation, we introduce an additional token to each node by aggregating the embeddings of its assigned codebooks:

$$h_c^v = \sum_{i=1}^{c} \mathbf{C}[i, t_i^v] \tag{14}$$

We found that adding this explicit aggregated token leads to better performance compared to initializing $\mathbf{X}_T$ with $\mathbf{C}$. The input representation of the sequence for node $v$ is then defined as:

$$S_v = \left[ \mathbf{X}_T[i, t_i^v] \underset{i=1}{\overset{c}{\|}} h_c^v \| \mathbf{X}_T[i, t_i^{u_1}] \underset{i=1}{\overset{c}{\|}} h_c^{u_1} \| \cdots \| \mathbf{X}_T[i, t_i^{u_k}] \underset{i=1}{\overset{c}{\|}} h_c^{u_k} \right] \tag{15}$$

This representation combines the individual token embeddings with the aggregated codebook embeddings, providing a more comprehensive and nuanced input to the Transformer.

In order to provide the Transformer with the global structural importance scores of the nodes within the sequence with respect to the target node, we introduce a gating mechanism over the input token embeddings as follows:

$$S_v = S_v \odot \text{Softmax} \left( \text{topk} \, \text{PPR} \left( v, \mathcal{E} \cup \mathcal{E}_s \right) \right) \tag{16}$$

where we first apply a softmax function with temperature $\tau = 1$ to normalize the PPR scores, and then multiply each node token's representation by its corresponding normalized score.

We also introduce two trainable positional encodings to the input tokens. The first positional encoding enables the Transformer to distinguish between tokens from different nodes, while the second encoding, referred to as hierarchical encoding, allows the Transformer to recognize the hierarchy level of each token within the codebooks. We randomly initialize the positional encodings $\mathbf{PE} \in \mathbb{R}^{(k+1) \times d_x}$ and $\mathbf{HE} \in \mathbb{R}^{c \times d_x}$ and sum them with the encoding of their corresponding token. For example, the final encoding of the token $j$ of the node $i$ within the sequence is computed as: $x = \mathbf{X}_T[j, t_j^{u_i}] + \mathbf{PE}[i] + \mathbf{HE}[j]$. Note that we did not use any structural encoding, such as Laplacian eigenvectors, as our experiments did not show any significant benefits from including them.

## 5.3 TRANSFORMER ENCODER & CLASSIFICATION HEAD

We use $l$ layers of standard Transformer encoder with flash attention (Dao et al., 2022b) to generate contextual representations per token in the sequence: $\mathbf{H}^{(l)} \in \mathbb{R}^{(c+1) \times (k+1) \times d_h}$. We then aggregate the token representations for $j$-th node in the sequence by summing along the token dimension:

$$\mathbf{H}_{v_j} = \sum_{i=1}^{c+1} \mathbf{H}^{(l)}[i, j] \in \mathbb{R}^{(k+1) \times d_h} \tag{17}$$

To obtain a single representation for the entire sequence, We further aggregate the representation using a linear attention layer:

$$h = \sum_{i=1}^{k+1} \alpha_i h_i \quad \text{where} \quad \alpha_i = \frac{\exp(\mathbf{W} h_i)}{\sum_j \exp(\mathbf{W} h_j)} \tag{18}$$

We feed the resulting representation into a fully-connected classifier and train the model end-to-end with cross-entropy loss. Note that during inference, only the Transformer and classifier are utilized, as the tokenizer is pretrained and the sequences are pre-computed. Furthermore, since we only require discrete tokens and codebook embeddings, our approach allows for efficient memory usage, regardless of graph size enable efficient training and inference on large-scale graphs.

## 6 EXPERIMENTS

We comprehensively evaluate GQT on both medium-scale and large-scale node classification tasks, encompassing both homophilous and heterophilous settings across 18 datasets. Homophilous graphs

Table 1: Mean node classification performance on medium-scale homophilous datasets over five runs.

| | | CoraFull | CiteSeer | PubMed | Computer | Photo | CS | Physics | WikiCS |
|---|---|---|---|---|---|---|---|---|---|
| **Dataset** | #Nodes | 19,793 | 3,327 | 19,717 | 13,752 | 7,650 | 18,333 | 34,493 | 11,701 |
| | #Edges | 126,842 | 4,522 | 88,651 | 491,722 | 238,163 | 163,788 | 495,924 | 216,123 |
| | #Features | 8,710 | 3,703 | 500 | 767 | 745 | 6,805 | 8,415 | 300 |
| | #Classes | 70 | 6 | 3 | 10 | 8 | 15 | 5 | 10 |
| | Measure | Accuracy ↑ | Accuracy ↑ | Accuracy ↑ | Accuracy ↑ | Accuracy ↑ | Accuracy ↑ | Accuracy ↑ | Accuracy ↑ |
| **GNN** | GCN | 61.76±0.14 | 76.50±1.36 | 86.54±0.12 | 89.65±0.52 | 92.70±0.20 | 92.92±0.12 | 96.18±0.07 | 77.47±0.85 |
| | GAT | 64.47±0.18 | 76.55±1.23 | 86.32±0.16 | 90.78±0.13 | 93.87±0.11 | 93.61±0.14 | 96.17±0.08 | 76.91±0.82 |
| | APPNP | 65.16±0.28 | 76.53±1.16 | 88.43±0.15 | 90.18±0.17 | 94.32±0.14 | 94.49±0.07 | 96.54±0.07 | 78.87±0.11 |
| | GPRGNN | 67.12±0.31 | 77.13±1.67 | 89.34±0.25 | 89.32±0.29 | 94.49±0.14 | 95.13±0.09 | 96.85±0.08 | 78.12±0.23 |
| | GraphSAINT | 67.85±0.21 | – | 88.96±0.16 | 90.22±0.15 | 91.72±0.13 | 94.41±0.09 | 96.43±0.05 | – |
| | GraphSAGE | – | 75.58±1.33 | 87.48±0.38 | 91.20±0.29 | 94.59±0.14 | 93.91±0.13 | 96.49±0.06 | 74.77±0.95 |
| | PPRGo | 63.54±0.25 | – | 87.38±0.11 | 88.69±0.21 | 88.69±0.12 | 92.52±0.15 | 95.51±0.08 | 78.12±0.23 |
| | GRAND+ | 71.37±0.11 | – | 88.64±0.09 | 88.74±0.11 | 94.75±0.12 | 93.92±0.08 | 96.47±0.04 | – |
| **GT** | GT | 61.05±0.38 | – | 88.79±0.12 | 91.18±0.17 | 94.74±0.13 | 94.64±0.13 | 97.05±0.05 | – |
| | Graphormer | OOM | – | OOM | OOM | 92.74±0.14 | 94.64±0.13 | OOM | – |
| | SAN | 59.01±0.34 | – | 88.22±0.15 | 89.93±0.16 | 94.86±0.10 | 94.51±0.15 | OOM | – |
| | GraphGPS | 55.76±0.23 | 76.99±1.12 | 88.94±0.16 | OOM | 95.06±0.13 | 93.93±0.15 | OOM | 78.66±0.49 |
| | GOAT | – | 76.89±1.19 | 86.87±0.24 | 90.96±0.90 | 92.96±1.48 | 94.21±0.38 | 96.24±0.24 | 77.00±0.77 |
| | NodeFormer | – | 76.33±0.59 | 89.32±0.25 | 86.98±0.62 | 93.46±0.35 | 95.64±0.22 | 96.45±0.28 | 74.73±0.94 |
| | DIFFormer | – | 76.72±0.68 | 89.51±0.67 | 91.99±0.76 | 95.10±0.47 | 94.78±0.20 | 96.60±0.18 | 73.46±0.56 |
| | NAGphormer | 71.51±0.13 | 77.42±1.41 | 89.70±0.19 | 91.22±0.14 | 95.49±0.11 | 95.75±0.09 | 97.34±0.03 | 77.16±0.72 |
| | Exphormer | 69.09±0.72 | 76.83±1.24 | 89.52±0.54 | 91.59±0.31 | 95.27±0.42 | 95.77±0.15 | 97.16±0.13 | 78.54±0.49 |
| | VCR-Graphormer | 71.67±0.10 | – | 89.77±0.15 | 91.75±0.15 | **95.53±0.14** | 95.37±0.04 | 97.34±0.04 | – |
| | GQT (ours) | **71.81±0.21** | **77.84±0.94** | **90.14±0.16** | **92.05±0.16** | 95.35±0.18 | **96.11±0.09** | **97.53±0.06** | **79.65±0.52** |

are characterized by nodes with similar classes being connected to each other, whereas heterophilous graphs exhibit connections between nodes with different classes. Following the convention of most existing works on GTs, we focus on node classification tasks in our experiments. However, as discussed in Section 3, our model can be easily extended to graph classification and link prediction tasks. For each evaluation scenario, we adhere to the established experimental protocols from previous works to ensure fair comparisons. Detailed descriptions of the datasets are provided in Appendix A and detailed experimental setup and hyperparameters are provided in Appendix B.

### 6.1 COMPARISON WITH STATE-OF-THE-ART

**Homophilous Node Classification.** To evaluate the performance on medium-scale homophilous graphs, we use eight benchmark datasets including CoraFull (Bojchevski & Günnemann, 2017), CiteSeer, and PubMed (Yang et al., 2016), Amazon Computers, Amazon Photos, Co-author CS, and Co-author Physics (Shchur et al., 2018), as well as WikiCS (Mernyei & Cangea, 2020). We compare our results with a comprehensive set of baselines, including four traditional GNNs: GCN (Kipf & Welling, 2017), GAT (Veličković et al., 2019), APPNP (Gasteiger et al., 2018), and GPRGNN (Chien et al., 2020); four scalable GNN variants including GraphSAINT (Zeng et al., 2019), GraphSAGE (Hamilton et al., 2017b), PPRGo (Bojchevski et al., 2020), and GTAND+(Feng et al., 2022); four standard GTs including GT (Dwivedi & Bresson, 2020), Graphormer (Ying et al., 2021b), SAN (Kreuzer et al., 2021), and GraphGPS (Rampášek et al., 2022); and six state-of-the-art scalable GTs including GOAT (Kong et al., 2023a), NodeFormer (Wu et al., 2022a), DiffFormer (Wu et al., 2023a), NAGphormer (Chen et al., 2023), Exphormer (Shirzad et al., 2023), and VCR-Graphormer (Fu et al., 2024). The baseline performance is reported from existing works (Wu et al., 2023b; Luo et al., 2024a; Fu et al., 2024). As shown in Table 1, GQT outperforms the baseline GNN and GT models on 7 out of 8 benchmarks. Notably, this achievement comes with a significant reduction in memory requirement for node features during Transformer training and inference. For example, on the Physics dataset with 34,493 nodes, we only use $256 \times 6$ tokens, i.e., 23-fold memory reduction.

**Heterophilous Node Classification.** Furthermore, we evaluate GQT on six small or medium-scale heterophilous datasets: Squirrel and Chameleon (Rozemberczki et al., 2021), Questions, Roman-Empire, Amazon-Ratings, and Minesweeper (Platonov et al., 2023b). We compare the performance with seven variants of GNNs including GCN, GraphSAGE, GAT, GPRGNN, H2GCN (Zhu et al., 2020a), CPGNN (Zhu et al., 2021), and GloGNN (Li et al., 2022), and six variants of GTs, including GraphGPS, GOAT, NodeFormer, SGFormer, NAGphormer, and Exphormer. The baseline performance is reported from existing works (Wu et al., 2023b; Luo et al., 2024b; Platonov et al.,

Table 2: Mean node classification performance on heterophilous graphs over five runs.

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

## 7 CONCLUSION

We introduced GQT (**G**raph **Q**uantized **T**okenizer) to decouple graph tokenization from Transformer using multi-task graph self-supervised learning. The GQT uses vector quantization to learn hierarchical tokens, resulting in significantly reduced memory requirements and improved generalization. We also introduced structural gating, hierarchical encoding, and semantic edges to further improve the performance. We achieved state-of-the-art performance on 16 out of 18 datasets, including large-scale homophilic and heterophilic datasets, while significantly reducing memory requirements. As future directions, we plan to explore the effectiveness of the GQT in graph generative learning by transitioning to a Transformer decoder. Our research lays the groundwork for further investigation into Graph Foundational Models, where LLMs can project heterogeneous features from diverse datasets into a unified textual representation. Building on this foundation, our GQT model can then convert a large number of nodes across different datasets into an efficient set of tokens.

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

# Appendix

## A   DATASETS

Here we provide a detailed description of the datasets we used. All datasets are publicly available.

- **CoraFull** (Bojchevski & Günnemann, 2017), **CiteSeer**, and **Pubmed** (Namata et al., 2012) are citation datasets, where nodes represents documents and edges represent citation link. Labels indicates paper category.

- **Computer** and **Photo** (Shchur et al., 2018) are from Amazon co-purchase graph (McAuley et al., 2015), where nodes represent goods and edges indicate that two goods are frequently bought together. Node features are bag-of-words encoded product reviews, and class labels are given by the product category.

- **CS** and **Physics** (Shchur et al., 2018) are co-authorship graphs based on the Microsoft Academic Graph from the KDD Cup 2016 challenges. Here, nodes are authors, that are connected by an edge if they co-authored a paper; node features represent paper keywords for each author's papers, and class labels indicate most active fields of study for each author.

- **WikiCS** (Mernyei & Cangea, 2020) is derived from Wikipedia, where nodes are Computer Science articles, and edges are based on hyperlinks. Nodes are classified into 10 classes representing different branches of the field.

- **Squirrel** and **Chameleon** (Rozemberczki et al., 2021; Pei et al., 2020) are Wikipedia page-page networks, where nodes represent articles from the English Wikipedia, and edges reflect mutual links between them. The nodes were classified into 5 classes in terms of their average monthly traffic.

- **Amazon-Ratings** (Platonov et al., 2023b) is based on the Amazon product co-purchasing data. Nodes are products (books, music CDs, DVDs, VHS video tapes), and edges connect products that are frequently bought together. The task is to predict the average rating given to a product by reviewers.

- **Roman-Empire** (Platonov et al., 2023b) is based on the Roman Empire article from English Wikipedia. Each node in the graph corresponds to one (non-unique) word in the text. Thus, the number of nodes in the graph is equal to the length of the article. Two words are connected if these words follow each other in the text, or these words are connected in the dependency tree of the sentence. The class of a node is its syntactic role.

- **Minesweeper** (Platonov et al., 2023b) is inspired by the Minesweeper game. The graph is a regular 100x100 grid where each node (cell) is connected to eight neighboring nodes (with the exception of nodes at the edge of the grid, which have fewer neighbors). 20% of the nodes are randomly selected as mines. The task is to predict which nodes are mines. The node features are one-hot-encoded numbers of neighboring mines. However, for randomly selected 50% of the nodes, the features are unknown, which is indicated by a separate binary feature.

- **Questions** (Platonov et al., 2023b) is based on data from the question-answering website Yandex Q, where nodes are users, and an edge connects two nodes if one user answered the other user's question during a one-year time interval (from September 2021 to August 2022). The task is to predict which users remained active on the website, forming a binary classification task.

- **ogbn-proteins** (Hu et al., 2020a) is a protein-protein assiciation network, where nodes represent proteins, and edges indicate different types of biologically meaningful associations between proteins, e.g., physical interactions, co-expression or homology. The task is to predict the presence of protein functions in a multi-label binary classification setup.

- **ogbn-arxiv** (Hu et al., 2020a) is a citation network between all Computer Science (CS) arXiv papers indexed by MAG (Wang et al., 2020). Each node is an arXiv paper and each directed edge indicates that one paper cites another one. The task is to predict the 40 subject areas of arXiv CS papers, e.g., cs.AI, cs.LG, and cs.OS.

- **ogbn-products** (Hu et al., 2020a) is an Amazon product co-purchasing network[1] of 2M products. Edges indicate that the products are purchased together. The task is to predict the category of a product.
- **pokec** (Leskovec & Krevl, 2014; Lim et al., 2021) is a social network, where nodes are users, and edges represent friendships. The task is to predict the gender of users.

For CoraFull, Pubmed, PubMed, Computer, Photo, CS, and Physics, we follow previous work and use 60%/20%/20% train/valid/test split. For WiKiCS, we follow the official split in Mernyei & Cangea (2020). For Squirrel, Chameleon, Amazon-Ratings, Roman-Empire, Minesweeper, and Questions, we follow the splits in Platonov et al. (2023b). For ogbn-proteins, ogbn-arxiv, and ogbn-papers, we follow the splits in Hu et al. (2020a). And for pokec, we follow the split used in Lim et al. (2021).

## B  EXPERIMENTAL SETUP

**Software and hardware.** The implementation of our method is based on PyTorch[2], PyG[3], DGL[4], and vector-quantize-pytorch package[5]. Most of the datasets can be accessed from PyG and DGL. All the experiments are conducted on one Nvidia A100 GPU.

**Hyperparameters and experimental details.** As illustrated in Figure 1, our method includes two parts: tokenizer and Transformer. We provide the hyperparameters and experimental details for each parts below.

During the training of graph tokenizer, we use full-graph training for small and medium-scale datasets, and apply sampling for large-scale graphs. We consider different sampling methods including random partitioning which randomly samples nodes within a graph and returns their induced subgraph, neighbor sampling (Hamilton et al., 2017b), GraphSAINT (Zeng et al., 2019), and local clustering used in Hou et al. (2023). For the GNN encoder and decoder, we use GCN or GAT as our backbone and tune the number of layers from $\{1, 2, 3, 4, 5, 6, 7, 8, 9, 10\}$ and hidden dimensions from $\{128, 256, 512, 1024\}$. For the quantizer, we use residual-VQ (RVQ) (Lee et al., 2022) and tune the number of codebooks from $\{1, 2, 3, 6, 9\}$ and codebook size from $\{128, 256, 512, 1024\}$. We set the code dimension to be the hidden dimension of the GNN encoder.

During the training of Transformer, we use KNN to add semantic edges and tune the number of semantic neighbors from $\{0, 5, 10, 15, 20\}$. Then we use PPR to generate a sequence of nodes for each target node. We tune the number of PPR neighbors from $\{0, 5, 10, 20, 30, 50\}$. For the Transformer model, we use the TransformerEncoder module in PyTorch ad our backbone, and tune the number of layers from $\{1, 2, 3, 4, 5, 6\}$, number of heads from $\{4, 8\}$, and feedforward dimension from $\{512, 1024, 2048\}$.

## C  FURTHER ABLATION STUDY

Additionally, we provide ablation study on one of the heterophilous dataset. Results are shown in Table 5. Results show that introducing semantic edges and structural gating mechanisms specifically benefits the heterophilous setting.

---

[1]http://manikvarma.org/downloads/XC/XMLRepository.html
[2]https://pytorch.org/
[3]https://pyg.org/
[4]https://www.dgl.ai/
[5]https://github.com/lucidrains/vector-quantize-pytorch

Table 5: Ablation study on effect of proposed components on the Minesweeper dataset.

| | Graph Tokenizer | | | Token Modulation | | | Augmentation | | Model | Performance |
|---|---|---|---|---|---|---|---|---|---|---|
| | RVQ | GMAE2 | DGI | Codebook Embeddings | Positional Encoding | Structural Gating | Semantic Edges | PPR Sequence | | ROC-AUC$\uparrow$ |
| (1) | ✓ | ✓ | ✓ | ✓ | | | | | Linear | 90.11 |
| (2) | | | | | ✓ | | | ✓ | Transformer | 90.65 |
| (3) | | ✓ | ✓ | ✓ | ✓ | ✓ | ✓ | ✓ | Transformer | 95.33 |
| (4) | ✓ | | ✓ | ✓ | ✓ | ✓ | ✓ | ✓ | Transformer | 92.86 |
| (5) | ✓ | ✓ | | ✓ | ✓ | ✓ | ✓ | ✓ | Transformer | 93.85 |
| (6) | ✓ | ✓ | ✓ | | ✓ | ✓ | ✓ | ✓ | Transformer | 93.12 |
| (7) | ✓ | ✓ | ✓ | ✓ | | ✓ | ✓ | ✓ | Transformer | 94.89 |
| (8) | ✓ | ✓ | ✓ | ✓ | ✓ | | ✓ | ✓ | Transformer | 93.97 |
| (9) | ✓ | ✓ | ✓ | ✓ | ✓ | ✓ | | ✓ | Transformer | 92.45 |
| (10) | ✓ | ✓ | ✓ | ✓ | ✓ | ✓ | ✓ | ✓ | Transformer | 95.28 |