# OpenReview forum: "Learning Graph Quantized Tokenizers"
_ICLR.cc/2025/Conference — ICLR 2025 Poster_

### Official Review · Reviewer_D7qL · 2024-11-03

**Soundness:** 3
**Presentation:** 3
**Contribution:** 4
**Rating:** 8
**Confidence:** 4

**Summary:**

In this work, the authors propose GQT, a transformer-based framework for graph prediction tasks. GQT consists of a variety of proposed components, including a custom training loss and a tokenizer trained separately with a self-supervised learning approach. The tokenizer leverages vector quantization, resulting in a highly efficient tokenization that can be fed into a standard transformer encoder and applied to very large graphs with up to 2.5 million nodes. Empirically, GQT outperforms a variety of GNN and transformer baselines on 16 out of 18 transductive node classification problems, including both homophilic and heterophilic graphs.

**Strengths:**

- **S1**: The authors evaluate their model on a wide variety of transductive node classification datasets, both homophilic and heterophilic and up to 2.5 million nodes. The empirical results are impressive and indicate that GQT is a versatile approach to transductive node classification.
- **S2**: The authors present an ablation study in Table 4 and Table 5 on the various components of their approach. The study indicates that most components help with improving the overall model performance; see **Q1** for a follow-up question on the ablation study.
- **S3**: The approach for learning a tokenizer using vector quantization is highly interesting and should be further explored in future research. In particular, I expect that GQT can be readily adapted to other types of graph learning problems (such as inductive settings) while being highly efficient.

**Weaknesses:**

- **W1**: While the results are impressive, the authors evaluate exclusively on transductive node classification problems. In my opinion, the authors should be more upfront about this limitation or include results for inductive settings, e.g., the inductive tasks in [https://arxiv.org/abs/2206.08164](https://arxiv.org/abs/2205.12454).
- **W2**: Section 4.2 introduces the training loss with three different loss terms. However, the authors do not provide any intuition or justification for these choices. As such, it is difficult for a reader to understand why these choices were made and how crucial they are to the overall framework. A similar issue appears in Sections 5.1 and 5.2.
- **W3**: In the introduction in L35-36, the authors state that “GTs possess an expressive power equivalent to the 2-Weisfeiler-Lehman (WL) isomorphism test”, citing Kim et al. (https://arxiv.org/abs/2207.02505).  The transformer used in Kim et al. is entirely different from the transformers used in transductive node classification and in particular, different from GQT. As such, this reference does not serve as a motivation to the approach present in this work. Further, note that the GQT uses a GNN to learn local interactions and would suffer from the same expressivity limitations as GNNs, leading to the next statement in L37-38: “This [the 2-WL expressivity of GTs] surpasses the expressive power of message-passing GNNs, which are limited to the 1-WL test”. This statement is incorrect. The 1-WL and 2-WL tests as defined in Kim et al. are equivalent in expressive power. Finally, in L38-39, the authors state that “[…] a Transformer with sufficient attention heads can match or exceed the expressive power of a second-order invariant graph network, outperforming message-passing GNNs”, again citing Kim et al. Here, the authors omit the fact that transformers can only achieve said result when receiving as input $O(n^3)$ number of tokens, resulting in a transformer with $O(n^6)$ runtime and memory complexity. Again, this approach is entirely different to GQT and does not serve as a motivation for GQT. I strongly recommend the authors to adjust the introduction, remove the incorrect claims and provide a proper motivation for their approach.

**Questions:**

- **Q1**: Which components of GQT are also applicable to baselines? For example, while the tokenizer is certainly unique to GQT, adding semantic edges and PPR features to nodes may also improve baseline performance. I wonder whether the authors can give any indication as to whether the improved performance of GQT stems largely from components unique to GQT or whether their proposed components may be useful in improving the results of transformers for transductive node classification in general.
- **Q2**: Table 4 and 5: Are the standard deviations missing from these tables or was the ablation study performed only on a single seed? In case the study was performed on a single seed, I strongly recommend to repeat the ablation study for at least three seeds to verify that the results are statistically significant.
- **Q3**: Figure 1, the experimental section, the ablation studies as well as the conclusion mention “structural gating”. However, this term is never formally introduced. I suppose the authors refer to the gating mechanism in Equation 16. If this is the case, I encourage the authors to define properly define the term there.

---

> ### Author Response · Authors · 2024-11-22
> **Response to Reviewer D7qL**
>
> Thank you very much for the constructive feedback. Please find our responses below.
>
> **Other Graph Tasks**
>
> Please refer to our new experiments discussed in the **[general response](https://openreview.net/forum?id=oYSsbY3G4o&noteId=Au8CNrOBD3)**  to this question. In summary, we have conducted new experiments on a diverse set of graph tasks including graph classification and regression, inductive node classification, and inductive link prediction on long-range graph benchmarks demonstrating the strong performance of our model.
>
> **Training Losses**
>
> We have updated the paper to address this point. The reason we used three SSL objectives in a multi-task manner is to ensure that we capture different aspects of information by using different high-performing families of SSL objectives: student-teacher distillation, masked autoencoding, and Infomax. We also cite two papers that show this is indeed the case (Lines 203-204). As we show empirically, these objectives each contribute to the overall performance.
>
> **WL Test**
>
> Thank you for your comments. In our initial version, we included these statements to highlight the significant expressive power (theoretically) of Graph Transformers (GTs), which is one of our motivations for focusing on GTs in this paper. We agree that the transformer used in Kim et al. differs from the one in our GQT, and the statements made were not entirely appropriate. As you suggested, we have updated the paper and uploaded the new version. In the new introduction, we analyze the pros and cons of GNNs and GTs, highlighting the unique advantage of combining them, i.e., GNNs can provide locality information into tokens such that GTs do not need to perform dense pair-wise attention but only rely on a few tokens to learn the long-range information of graphs, which is the motivation for our proposed GQT.
>
> **Transferable Components of GQT**
>
> The contribution of semantic edges can be inferred from Tables 4 and 5. Specifically, by keeping everything intact and just dropping semantic edges, we observe a sharper downstream performance degradation in the heterophily setting (Table 5) compared to the homophily setting (Table 4). This indicates that introducing semantic edges helps in both scenarios but is more important for the heterophily setting. We speculate that this can be considered as a general augmentation even for GNNs improving their performance. For PPR, previous works have already shown that using PPR by replacing the adjacency matrix [1] improves GNNs, and also when used as positional encoding [2] or when used to generate sequences [3] can improve the performance of Transformers. Also Tables 4 and 5 suggest that using PPR scores (structural gating) can contribute to the performance of the model. This is also transferable to baselines. We will add more experiments to the camera-ready version comparing PPR sequences with BFS, DFS, and other random-walk based sequences.
>
> **Missing Standard Deviations in Tables 4 and 5**
>
> Thanks for catching this; we have updated them in the paper.
>
> **Structural Gating**
>
> Yes, that is correct. We introduce structural gating in Lines 343-349. We have updated the paper to make it clearer.
>
> Once again, we would like to thank you for the helpful suggestions, and we look forward to addressing any other questions you may have.
>
> **References**
>
> [1] Gasteiger et al. Diffusion Improves Graph Learning. NeurIPS 2019
>
> [2] Khasahmadi et al. Memory-Based Graph Networks. ICLR 2019.
>
> [3] Fu et al. VCR-graphormer: A mini-batch graph transformer via virtual connections. ICLR 2024.

---

> > ### Comment · Reviewer_D7qL · 2024-11-23
> >
> > I want to thank the authors for addressing my concerns. The new inductive task results look very promising. Thank you also for providing a more nuanced motivation in the introduction.

---

> > > ### Author Response · Authors · 2024-11-23
> > >
> > > Thank you. We would love to address any further questions you may have or clarify any part that is unclear to increase your confidence in your evaluation and in our work.

---

### Official Review · Reviewer_FMi4 · 2024-11-04

**Soundness:** 2
**Presentation:** 3
**Contribution:** 2
**Rating:** 5
**Confidence:** 5

**Summary:**

This work attempt to design a graph tokenizer that utilizes graph self-supervised learning to train a encoder. And then leverage the Transformer to capture long-range dependencies. Specifically, the authors adapts Residual Vector Quantization to learn hierarchical discrete tokens for reducing memory requirements and improving generalization capabilities.

**Strengths:**

1. The proposed Graph Quantized Tokenizer utilizes discrete tokens, which can significantly reduce memory requirements.

**Weaknesses:**

1. The major concern of this manuscript is the experiment part. According to [1], classic GNNs (e.g. GCN, GraphSAGE and GAT) can also achieve excellent performance on node classification tasks. However, the proposed GQT obtains lower performance on almost all datasets. This makes me worry about the actual contribution of this work.

2. The novelty of the proposed method is limited. There are already some works try to utilize vector quantization to learn discrete tokens for graph learning [2,3]. And there are also existing work utilize graph augmentation and Personalized PageRank to generate a sequence per node [4].

3. There are also some concept errors in the manuscript. For example, the authors mention that "Recent studies on applying Large Language Models (LLMs) to graph-related tasks have found the representing ..." in the introduction section. But most recent methods about combining LLMs with graph learning can only work on textual-attributed graph, and they just leverage LLMs to capture better textual  understanding [5, 6].


[1] Luo, Yuankai, Lei Shi, and Xiao-Ming Wu. "Classic GNNs are Strong Baselines: Reassessing GNNs for Node Classification." arXiv preprint arXiv:2406.08993 (2024).

[2] Yang, Ling, et al. "VQGraph: Rethinking graph representation space for bridging GNNs and MLPs." The Twelfth International Conference on Learning Representations. 2024.

[3] Xia, Jun, et al. "Mole-bert: Rethinking pre-training graph neural networks for molecules." (2023).

[4] Fu, Dongqi, et al. "VCR-graphormer: A mini-batch graph transformer via virtual connections." arXiv preprint arXiv:2403.16030 (2024).

[5] Li, Yuhan, et al. "A survey of graph meets large language model: Progress and future directions." arXiv preprint arXiv:2311.12399 (2023).

[6] Jin, Bowen, et al. "Large language models on graphs: A comprehensive survey." IEEE Transactions on Knowledge and Data Engineering (2024).

**Questions:**

1. Could the authors provide the explanation about why GQT obtains lower performance than classic GNNs?

2. I think the authors need to re-emphasize the innovation of this work.

---

> ### Author Response · Authors · 2024-11-22
> **Response to Reviewer FMi4 - Part 1**
>
> We would like to thank the reviewer for their constructive feedback. Please find our responses below:
>
> **Performance Comparison with Classic GNN Paper**
>
> Thanks for pointing this interesting paper which provides a thorough empirical analysis of classic GNNs.
> We did not report the results from [1] due to two main reasons: (1) we, along with several other groups, have been unable to reproduce the results reported in [1] using their exact setting. Therefore, we were cautious about reporting that work before validating it. (2) Moreover, we reported the performance of GNNs from their original papers, as our goal was not to perform extensive hyperparameter tuning, unlike [1]. This is also the case with other Graph Transformer works that are consistently outperformed by [1] (e.g., Table 2 in [1]). Nonetheless, to address your concerns, we performed two sets of experiments. (1) Reproducing the results from [1] using their exact setting, and (2) performing the same hyperparameter tuning for the GNN encoder of the tokenizer. We emphasize that we did not perform hyperparameter tuning for the Transformer encoder.  The results are shown below. As shown, (1) the reproduction results are inferior to those reported in [1], and (2) with the same hyperparameter tuning only on the GNN encoder of the tokenizer, we consistently outperform the results we obtained from their method.
>
> | | Computer | wikiCS | Squirrel | Minesweeper |
> |---|---|---|---|---|
> | Classic GNN (GAT) (reported) [1] | 94.09±0.37 | 81.07±0.54 |41.73±2.07 |97.73±0.73|
> | ClassicGNN (GAT) (reproduced) | 92.82±0.23 | 79.38±0.78 | 40.81±1.16 |96.52±0.47|
> | Ours (reported) | 92.05±0.16 | 79.65±0.52 | 42.54±1.37 |95.28±0.44|
> | Ours (further tuned) |  93.37±0.44 |  80.14±0.57 |  42.72±1.69| 96.70±0.38|
>
> **LLMs**
>
> We agree that current LLMs for graph tasks are mostly applied to textual-attributed graphs (TAGs). What we meant here was to focus on LLM-based methods that directly take the structure into account and perform aggregation in in-context or struct-tuning settings; [2-3] are examples of such approaches. We rewrote the paragraph and updated the citations to make it clearer.
>
> **Innovation**
>
> We already cite VQGraph [4] in the paper (Lines 121-123), which differs from our approach as they use a VQ codebook (we use RQ) for distilling GNNs to MLPs. Thank you for suggesting Mole-bert [5]; we missed this one as it is specific to molecules whereas our method is domain-agnostic. We added this work to the updated literature review. The differences between Mole-bert and our approach are as follows: (1) unlike Mole-bert, which is specifically designed for molecules, our work is general and domain-agnostic,  (2) Mole-bert uses VQ along with a GNN to perform BERT-style pretraining, whereas we only use the GNN module for tokenization and use a Transformer to learn long-range dependencies. This makes our proposed method more flexible, as it follows the standard tokenizer-transformer paradigm and is compatible with rapid advances in Transformer improvements;  (3) similar to VQGraph, Mole-bert uses VQ whereas we use RQ, which yields hierarchical codebooks. We ran an experiment to show the difference between using VQ and RQ on downstream tasks. The results shown below indicate that RQ significantly outperforms VQ due to its hierarchical nature.
>
> | | RQ | VQ |
> |--|--|--|
> | ogbn-arxiv | 73.14±0.16 | 72.05±0.18 |
> | Minesweeper | 95.28±0.44 | 91.02±0.48 |
>
>
> We also discussed VCR-Graphormer [7] in our model design and compared the performance in the experiments. In addition to our performance being better than VCR-Graphormer, our proposed tokenization is more advanced and novel than VCR-Graphormer for the following aspects:
>
> **Before sampling**. Our token obtaining is semantic, hierarchical, and learned by the residual vector quantization method, which means (1) our token is shorter than the original input feature and more friendly to the attention mechanism, as the relationship between attention computation cost and feature length is beyond linear; (2) our token is more friendly to the inductive setting, because VCR-Graphormer’s tokenization is _heuristic_ and needs to establish all prerequisites, such as identifying structure-aware and content-aware virtual connections and neighbors from scratch, whereas ours employs a trained tokenizer encoder, which can obtain token representations for new nodes in a one-shot manner without needing any rerunning.
>
> **After sampling**. VCR-Graphormer takes the tokens individually to feed them into the attention mechanism, which increases attention costs. Our attention mechanism makes the following novel efforts: (1) for attention efficiency, we take the summarization intra-codebooks to save attention complexity; (2) for attention effectiveness, we introduce trained embedding matrix to attach to each codebook for the output expressiveness, since our number of codebooks is only 3, which operation will not induce too many computational costs.

---

> > ### Author Response · Authors · 2024-11-22
> > **Response to Reviewer FMi4 - Part 2**
> >
> > To sum up, compared with VCR-Graphmorer, only the sampling method, i.e., personalized PageRank, is similar and that is a classic sampling method in the graph research community. Specifically, the entities being sampled are different and the computation after sampling is also different, which are deliberately upgraded for efficiency and effectiveness.
> >
> > Thanks once again for the feedback and we look forward to addressing more questions to strengthen the paper for the camera-ready version.
> >
> > **References**
> >
> > [1] Luo et al. Classic GNNs are Strong Baselines: Reassessing GNNs for Node Classification. arXiv 2024
> >
> > [2] Ye et al. Language is All a Graph Needs. EACL 2024
> >
> > [3] Xu et al. Language Models are Graph Learners. arXiv 2024
> >
> > [4] Yang et al. VQGraph: Rethinking graph representation space for bridging GNNs and MLPs. ICLR 2024
> >
> > [5] Xia et al. Mole-bert: Rethinking pre-training graph neural networks for molecules. ICLR 2023
> >
> > [6] Fu et al. VCR-graphormer: A mini-batch graph transformer via virtual connections. ICLR 2024

---

> > ### Comment · Reviewer_FMi4 · 2024-11-22
> > **Response to authors**
> >
> > Thanks for your answering.
> >
> > However, I have tried to reproduce the results in [1], and I found that most of results in Table 3 can be reproduced. Could you please reproduce the results of GCN on heterophilous graphs and compare your model with it?
> >
> > [1] Luo, Yuankai, Lei Shi, and Xiao-Ming Wu. "Classic GNNs are Strong Baselines: Reassessing GNNs for Node Classification." arXiv preprint arXiv:2406.08993 (2024).

---

> ### Author Response · Authors · 2024-11-22
> **Response to Reviewer FMi4**
>
> Thanks for your prompt response! We tuned on **GAT** since we used GAT as the graph encoder in our GQT. We selected Computer, wikiCS, Squirrel, Minesweeper, on which the tuned ClassicGNN (GAT) achieves largest improvement as reported in the paper [1] (we exclude Chameleon as the dataset size is very small).
>
> We also noticed that GCN in [1] achieves quite good results on Squirrel, even more than 7.8% improvement over the tuned SAGE and GAT, which is quite large on this long-used dataset.
>
> As you suggested, we will try to reproduce the results of GCN on **heterophilous graphs** and will update the results soon.
>
> [1] Luo, Yuankai, Lei Shi, and Xiao-Ming Wu. "Classic GNNs are Strong Baselines: Reassessing GNNs for Node Classification." arXiv preprint arXiv:2406.08993 (2024).

---

> ### Author Response · Authors · 2024-11-23
> **Response to Reviewer FMi4**
>
> Thanks again for pointing out this interesting paper, which provides a thorough empirical analysis of classic GNNs on the node classification task.
>
> 1. As you suggested, we tried to **reproduce the results of GCN** and **further tuned our GQT** on **heterophilous graphs**. The results are shown below. As shown, with additional hyperparameter tuning, we can outperform or be comparable to the results we obtained from their method.
>
>    We also listed the selected hyperparameters (# GNN layers, # GNN hidden dim) in the table. It can be observed that ClassicGNN typically requires a larger **number of layers** on these heterophilous graphs, which results in higher memory costs. In contrast, our GQT incorporates tokens learned from GNN (local) encoders and Transformer to capture long-range interactions, allowing us to use smaller GNN layers.
>
>    Note that the authors of [1] recently updated the code for these medium datasets, therefore, our new (reproduced) run is based on this code (previously, we followed the code in the “large_graph” folder).
>
> | | Amazon-Ratings | Roman-Empire | Minesweeper | Questions |
> |---|---|---|---|---|
> | ClassicGCN hyperparameters (# GNN layers, # GNN hidden dim) | 4, 512 | 9, 512 | 12, 64 | 10, 512 |
> | Ours hyperparameters (# GNN layers, # GNN hidden dim) | 4, 512 | 6, 512 | 6, 64 | 4, 512 |
> | ClassicGNN (reported) [1] | 53.80±0.60 | 91.27±0.20 |97.86±0.24 | 79.02±0.60 |
> | ClassicGNN (reproduced) | 53.63±0.33 | 90.92±0.11 | 97.47±0.06 | 78.38±0.11 |
> | Ours (reported) | 53.89±0.36 | 89.21±0.43 | 95.28±0.44 | 77.28±1.36 |
> | Ours (further tuned) |  54.32±0.41 |  90.98±0.24 |  97.36±0.35 |  78.94±0.86 |
>
> 2. In addition, we have included additional results on the **Long Range Graph Benchmark (LRGB)** [2] in [general response](https://openreview.net/forum?id=oYSsbY3G4o&noteId=Au8CNrOBD3) (including graph-level and link-level tasks), which is proposed to evaluate the capability of GNNs and GTs to capture long-range interactions. Our GQT (along with several other GT baselines) outperform GNNs on this benchmark, demonstrating the effectiveness of GTs (and our GQT) in capturing long-range dependencies.
>
> Additionally, we would like to emphasize that, our primary goal is to investigate an effective and transferable graph tokenization strategy that enables the learned tokens to encode local interactions, allowing a Transformer encoder to focus on long-range dependencies. Unlike [1], our objective was not to perform extensive hyperparameter tuning, which is also the case with other Graph Transformer works that are consistently outperformed by [1]. However, we acknowledge the contribution of [1] and its strong results. Nonetheless, as shown we either outperform or on par with their results. our To ensure a fair comparison, we performed additional tuning on our method.
>
> We sincerely thank you for your time! Hope we have addressed your concerns through the new experiments. We look forward to your reply and further discussions, thanks!
>
> [1] Luo, Yuankai, Lei Shi, and Xiao-Ming Wu. "Classic GNNs are Strong Baselines: Reassessing GNNs for Node Classification." arXiv preprint arXiv:2406.08993 (2024).
> [2] Dwivedi, Vijay Prakash, et al. "Long range graph benchmark." Advances in Neural Information Processing Systems 35 (2022): 22326-22340.

---

> > ### Comment · Reviewer_FMi4 · 2024-11-24
> > **Response to authors**
> >
> > Thanks for your answering. I have looked at your additional experimental results. They are comprehensive. And you have nicely addressed my concerns about novelty and concept errors. However, I found that the proposed GQT just achieves weak improvements compared with classicGNN. Given that classicGNN (e.g. GCN and GAT) are basic baselines in node classification task, I think these weak improvements are hard to verify the contribution of this work. And there are also some recent literatures about graph transformers achieve excellent performance on long range graph benchmarks [1].
> >
> > Based on what I have discussed, I will increase the Rating to 5.
> >
> > [1] Ma, Liheng, et al. "Graph inductive biases in transformers without message passing." International Conference on Machine Learning. PMLR, 2023.

---

> ### Author Response · Authors · 2024-11-24
>
> Thank you.
>
> In addition to presenting our model's strong performance across various benchmarks, we would like to take a step back and re-emphasize the broader significance of this paper. Two key aspects set our model apart from existing approaches:
>
> (1) Our model offers greater flexibility compared to specialized Graph Transformers. Unlike these models, our tokenizer can be seamlessly integrated into future advances in Transformer design, which is an area of research that is progressing at a much faster pace than Graph Transformers or GNNs. This versatility makes our design more adaptable to rapid progress in the field. When compared to GNNs, our approach eliminates the need to iterate through multiple architectures for each dataset to find the optimal GNN layer for a given task.
>
> (2) It is worth noting that all GNNs and specialized Graph Transformers require access to original node features during inference, which can be a significant limitation. In contrast, our design overcomes this challenge by learning discrete tokens. This is particularly important in large-scale industrial settings, where storing and retrieving node features for billions of users or items is not feasible due to memory constraints. We reported results comparing our model with a GAT model showing that with our design we can use a Transformer with half the memory footprint compared to GAT. This efficiency advantage is crucial in large-scale industrial applications.
>
> In summary, our model not only demonstrates strong performance across benchmarks but also offers two significant advantages: adaptability to rapid developments in the Transformer family of architectures and improved memory efficiency. We believe these contributions make our paper an important contribution to the community.
>
> We would be more than happy to address any additional questions you may have or provide further clarification on any aspects of our work to increase your confidence in your evaluation and in our work.

---

### Official Review · Reviewer_cy7L · 2024-11-04

**Soundness:** 2
**Presentation:** 2
**Contribution:** 3
**Rating:** 6
**Confidence:** 3

**Summary:**

This paper proposes a self-supervised method, named GQT, for graph token learning. GQT adopts the residual vector quantization to generate graph tokens and trains the model with mutual information maximization, graph reconstruction loss and distillation loss minimization. Given the learned codebook, GQT augments the input graph with edges between similar nodes and generates a node sequence for each node with topK-related nodes in personalized PageRank. Extensive results on node classification are provided to demonstrate the effectiveness of GQT.

**Strengths:**

- This paper decouples the graph tokenizer from transformer training, which poses an interesting thread for developing graph transformers.
- The proposed method achieves competitive performance on node classification, including heterophilic and homophilic datasets.
- GQT scales to benchmarks with large graphs.

**Weaknesses:**

- As the core module in GQT, the authors did not provide a sufficient description of the residual vector quantization (nor in the appendix). How does GQT learn the mapping from graph nodes to the codebook tokens? The mapping relation $X_Q$ is introduced in L194/199 and never used again in the following context.
- The writing of section 5.2 is rather arbitrary. I cannot fully comprehend the embedding matrix $X_T$, which is 'trained end-to-end with the Transformer'. How does $X_T$ be learned in GQT? In L333-334, the authors argue that 'adding this explicit aggregated token leads to better performance compared to initializing $X_T$ with $C$. Does this mean that $X_T$ is initialized with $C$ or not?
- The operator $[\cdot,\cdot]$ in Eq 14/15 lacks a definition.
- The GQT (or tokenizer) is claimed to (1) capture long-range dependencies; (2) less memory footprint; (3) improved generalization capabilities, which lack of supportive analysis in the experiments.

**Questions:**

-  How does GQT perform with different sizes of codebooks ($K$, $c$ and $d_c$).
- The graph codebook learning resembles node clustering. Can authors provide a comparison between these two types of methods? Or how does node clustering perform in the graph codebook learning?

---

> ### Author Response · Authors · 2024-11-22
> **Response to Reviewer cy7L - Part 1**
>
> We thank the reviewer for their positive response and constructive feedback. We address the raised questions as follows.
>
> **Vector Quantization**
>
> We have added an algorithm and detailed explanation of how graph nodes are mapped into tokens in the Appendix. A brief description is also provided in Section 3, Lines 153-161. In summary, the process involves passing the input graph through a GNN encoder to learn node representations. Each node representation is then compared against the embeddings of the first codebook, and the index of the closest embedding to each node’s representation is assigned to it as its first discrete token. The difference between the node representation and the assigned codebook embedding is then compared to the embeddings of the second codebook to find the second discrete token. This process repeats for all codebooks. $X_Q$ is a matrix of size $n \times c$, where n is the number of nodes and c is the number of codebooks. This matrix essentially holds the discrete tokens for nodes. For example, if we have 1000 nodes and 3 codebooks with 256 embeddings, each node will be represented by 3 discrete tokens where each token is an integer between 0-255. Thus, $X_Q \in \{0, 1, \cdots, 255\}^{1000 \times 3}$. $T_v$ in Line 296 is a row of this matrix that represents the 3 discrete tokens of node $v$. We have slightly changed the notation to make it clearer.
>
> **Embedding Matrix**
>
> The embedding matrix is not learned during training the tokenizer; instead, it is learned during training the Transformer. Following the example provided in the previous response, if we set the input dimension to the Transformer to d, then $X_T$ is a matrix of size $3 \times 256 \times d$, which is essentially the embedding table for discrete tokens. The embeddings are initialized randomly. As mentioned in Lines 333-334, we found that initializing this embedding table randomly and introducing a fourth token by summing up the embeddings of the codebooks for each node (Eq. 14) yields better results compared to initializing the embedding table with the embeddings of codebooks. We have updated the paper to clarify this point.
>
> **[.,.] Operator**
>
> Thank you for pointing out this. We have added a sentence to explain it in the updated paper.
>
> For Eq 14 $C[i, t_i^v]$, $i$ and $t_i^v$ are integers, and this operator is essentially the indexing operator on tensors. For example, $C[i, j]$ returns the embedding of the $j$-th entry in the $i$-th codebook.
>
> For Eq 15, we use [ || ] to show the concatenated sequence.
>
>
> **Supportive Analysis**
>
> We would like to clarify that we do not claim that the tokenizer captures long-range dependencies. Instead, we claim that the tokenizer uses a GNN to model local interactions, allowing the Transformer to focus on capturing long-range dependencies. This is analogous to Vision Transformers (ViTs) [1], where instead of overwhelming the Transformer with pixels, a linear or convolutional layer first models the local information on image patches. Nevertheless, we have conducted new experiments on **long-range graph benchmarks** to show that our model enables capturing such interactions. We have also addressed generalization and memory footprint in the **[general response](https://openreview.net/forum?id=oYSsbY3G4o&noteId=Au8CNrOBD3)**.
>
> **Different sizes of Codebooks**
>
> We treat the number of codebooks, codebook size, and embedding as hyperparameters and perform lightweight tuning for each dataset. As mentioned in the Appendix, we choose the number of codebooks from {1, 2, 3, 6, 9}, codebook size from {128, 256, 512, 1024}, and hidden dimensions from {128, 256, 512, 1024}. We have also added a Table to the appendix that reports the chosen hyperparameters for each dataset. To address your question, we have conducted experiments on ogbn-arxiv and Minesweeper datasets to investigate the effect of varying these hyperparameters. The results shown below suggest that as we increase each of these hyperparameters, the performance tends to increase first and then decrease, respectively. We will add more analysis to the camera-ready version.
>
> | | #Codebooks | Codebook size | Codebook dim | Performance |
> |---|---|---|---|---|
> | ogbn-arxiv (169,343 nodes) | 1 | 512 | 512 | 72.05±0.18 |
> | | 3 | 512 | 512 | 73.14±0.16 |
> | | 6 | 512 | 512 | 73.09±0.14 |
> | | 3 | 256 |  512  | 72.91±0.21 |
> | | 3 | 1024 |  512 |  73.05±0.15 |
> | | 3 | 512 |  256  | 72.96±0.19 |
> | | 3 | 512 |  1024 |  73.11±0.19 |
> | Minesweeper (10,000 nodes) | 1 | 128 | 128 | 91.02±0.48 |
> | | 3 | 128 | 128 | 95.28±0.44 |
> | | 6 | 128 | 128 | 95.11±0.56 |
> | | 3 | 64 | 128 | 93.87±0.51 |
> | | 3 | 256 | 128 | 95.23±0.39 |
> | | 3 | 128 | 64 | 94.96±0.49 |
> | | 3 | 128 | 256 | 95.04±0.53 |

---

> ### Author Response · Authors · 2024-11-22
> **Response to Reviewer cy7L - Part 2**
>
> **Node Clustering**
>
> Vector Quantization (VQ) is closely related to K-Means clustering, and the embeddings of the codebook can be interpreted as centroids of the clusters. A good heuristic for initializing the VQ codebook is, in fact, performing K-Means clustering on the first batch of data. However, unlike classic algorithms such as K-Means, VQ performs representation learning and clustering end-to-end. In this work, we used Residual Quantization (RQ) which is related to hierarchical clustering, where the embeddings of the first codebook can be interpreted as centroids of coarse-grained clusters, and the embeddings of the last codebook are centroids of fine-grained clusters. To investigate this further, we trained a variant of our tokenizer by removing the quantization layer. Once the model was trained, we clustered the codebooks using K-Means and used the cluster indices as tokens. The results are shown below. As the results suggest, RQ achieves better performance because clustering and representation learning are performed end-to-end. Meanwhile, the results confirm our findings in Table 4, indicating that multi-task SSL objectives have a more impact on downstream performance. Additionally, they indicate that the learned representations by the tokenizer are expressive, allowing us to achieve even good performance by clustering them using K-Means.
>
> | | RQ | K-Means |
> |--|--|--|
> | ogbn-arxiv | 73.14±0.16 |  72.35±0.21 |
> | Minesweeper | 95.28±0.44 | 92.48±0.56 |
>
> **References**
>
> [1] Dosovitskiy et al. An Image is Worth 16x16 Words: Transformers for Image Recognition at Scale. ICLR 2020.

---

> ### Comment · Reviewer_cy7L · 2024-11-25
>
> I thank the authors for their efforts during the rebuttal and for providing detailed responses. After reviewing the clarifications and additional explanations, my initial concerns have been addressed.
>
> However, I noticed that the authors have only added a comparison of inference time in the appendix. Given that GQT relies on a two-stage model training process, how does its training time compare to the single-stage GTs? Based on the current results, the performance improvement of GQT appears to be limited. It remains to be demonstrated whether the additional time cost for codebook training is justified.  Can the learned codebooks generalize to other datasets?
>
> Additionally, -*philous graphs* in the manuscript should be -*philic graphs*.

---

> > ### Author Response · Authors · 2024-11-26
> > **Response to Reviewer cy7L - Part 3**
> >
> > Thanks for the new questions, please find our responses as follows.
> >
> > **Training Time**
> > Thanks for the suggestion. We compared our training time with that of one GNN model and one Graph Transformer model on the Minesweeper dataset. Specifically, we employed GAT with hyperparameters as specified in [1] for the GNN, and VCR-Graphormer [2] for the Graph Transformer. The results are shown below.
> >
> > | | Total training time (Minesweeper) |
> > |--|--|
> > | GAT [1] | 3 mins |
> > | VCR-Graphormer [2] |  6 mins |
> > | Ours (Tokenizer + Transformer) |  1 + 2 mins |
> >
> > Our first stage (training the Tokenizer) is faster than training GAT mainly because our tokenizer requires fewer layers. Compared to VCR-Graphormer, we use fewer training epochs to converge. In addition, VCR-Graphormer requires extra preprocessing steps to precompute hop-aggregated embeddings and perform clustering. We will include training time comparisons for other datasets in the camera-ready version.
> >
> > **Performance Improvement**
> > As shown by the experiments in the paper and the subsequent experiments during rebuttal, our model exhibits strong performance, either surpassing or on par with strong baselines from various families of specialized models. Taking a step back, we would like to reiterate the broader significance of this work. Two key aspects set our model apart from existing approaches:
> >
> > (1) Our model offers greater flexibility compared to specialized Graph Transformers. Unlike these models, our tokenizer can be seamlessly integrated into future advances in Transformer design, which is an area of research that is progressing at a much faster pace than Graph Transformers or GNNs. This versatility makes our design more adaptable to rapid progress in the field (e.g., FlashAttention). When compared to GNNs, our approach eliminates the need to iterate through multiple architectures for each dataset to find the optimal GNN layer for a given task.
> >
> > (2) It is worth noting that all GNNs and specialized Graph Transformers require access to original node features during inference, which can be a significant limitation. In contrast, our design overcomes this challenge by learning discrete tokens. This is particularly important in large-scale industrial settings, where storing and retrieving node features for billions of users or items is not feasible due to memory constraints. We reported results comparing our model with a GAT model showing that with our design we can use a Transformer with half the memory footprint compared to GAT. This efficiency advantage is crucial in large-scale industrial applications.
> >
> > In summary, our model not only demonstrates strong performance across benchmarks but also offers two significant advantages: adaptability to rapid developments in the Transformer family of architectures and improved memory efficiency. We believe these contributions make our paper an important contribution to the community.
> >
> > **Codebook Generalize to Other Datasets**
> > Thanks for suggesting this interesting direction. Indeed, we can perform transfer learning across datasets that belong to the same domain and share the same input feature space (e.g., from molecules to molecules as in [3]). In this setting, we train the tokenizer on dataset A and then transfer it to dataset B. We conducted two experiments to explore this approach. Both experiments involved training the tokenizer on the ogbn-papers100M dataset and transferring it to the ogbn-arxiv dataset. The key difference between the two experiments lies in the treatment of the tokenizer: in one experiment, we froze the tokenizer, while in the second experiment, we performed a light fine-tuning with only 10 epochs on the ogbn-arxiv dataset. The results are shown below.
> >
> > | | Accuracy (ogbn-arxiv) |
> > |--|--|
> > | Tokenizer transferred from ogbn-papers with no fine-tuning | 72.36±0.24 |
> > | Tokenizer transferred from ogbn-papers with 10 epochs of fine-tuning |  73.21±0.21 |
> > | Tokenizer trained from scratch | 73.14±0.16 |
> >
> > Two key observations can be made: (1) even without fine-tuning, our model exhibits strong performance compared to other Graph Transformers (Table 3 in the paper), and (2) with light fine-tuning (only 10 epochs), the  model's performance surpasses that achieved by training the tokenizer from scratch, suggesting positive transfer.
> >
> > We would also like to highlight that our research lays the groundwork for further exploration into Graph Foundational Models where LLMs can project heterogeneous features from diverse datasets into a unified textual representation. Our GQT model can then convert a large number of nodes across different datasets into a compact set of tokens. This represents a promising future direction for our work. We will include more experiments on the transferability of the tokenizer in the camera-ready version.
> >
> > **--philous graphs to --philic graphs**
> > Thanks for the suggestion. We applied the change in the updated paper.

---

> > > ### Author Response · Authors · 2024-11-26
> > > **Response to Reviewer cy7L - Part 4**
> > >
> > > **References**
> > >
> > > [1] Classic GNNs are Strong Baselines: Reassessing GNNs for Node Classification, NeurIPS 2024
> > >
> > > [2] VCR-Graphormer: A Mini-batch Graph Transformer via Virtual Connections, ICLR 2024
> > >
> > > [3] Hu et al. Strategies for Pre-training Graph Neural Networks. ICLR 2020.

---

> > > > ### Comment · Reviewer_cy7L · 2024-11-26
> > > >
> > > > Thank you for the new results and discussion.
> > > >
> > > > The focus of this research offers valuable insights for improving the generalization capability of the GT family. Although the core components of the paper are derived from existing methods, the authors have made some effort in adapting them to make the method work.
> > > >
> > > > Based on the current responses, I will raise the score to 6. I hope the authors will incorporate the rebuttal content into the manuscript and further polish the writing. Good luck.

---

> ### Author Response · Authors · 2024-11-26
>
> Thank you for the constructive engagement. We will definitely update the paper based on the feedback we received in rebuttal period. Please let us know if you had any further questions and we are more than happy to answer them. Thanks again.

---

### Official Review · Reviewer_rTiq · 2024-11-04

**Soundness:** 2
**Presentation:** 2
**Contribution:** 2
**Rating:** 5
**Confidence:** 4

**Summary:**

The paper proposes Graph Quantized Tokenizer (GQT) that considers three key properties such as local interactions, memory efficiency, and robustness & generalization for graph Transformer. Different existing works that train tokenizer and Transformer architecture in an end-to-end manner, GQA decouples the training tokenizer from the Transformer. In other words, it first trains the tokenizer and then learns graph Transformer architecture given the tokenized graph. From the experimental results, GQA shows the best performance compared to other baselines on 7 out of 8 datasets. It also achieves the best performance on large-scale datsets.

**Strengths:**

- The design of tokenizers in graph Transformers is really an important research topic. The paper well describes the existing works of graph Transformers and tokenizers in the related works.
- The paper is easy to follow.
- The experiments of the paper demonstrate that the proposed method is effective with multiple experiments including homophilious, heterophilious and large-scale datasets.

**Weaknesses:**

- I think that the novelty of the paper is limited.
   - One of the main contributions highlighted in this paper is quantized tokenization for graph Transformers. However, the paper simply combines graph neural networks for dealing with graph-structured data and existing residual vector quantization for the quantization.
   - Another contribution is multi-task self-supervised learning. The paper uses two self-supervised learning objectives such as DGI and GMAE2, which are not proposed self-supervised training scheme in this paper. I think that using two self-supervised objectives to improve the effectiveness of the model is hard to become a good contribution in top-tier ML conferences.

- More experiments need to be included.
   - The author claims that the proposed quantization method improves the robustness and generalization of the model. But, there is no experimental result concerning the robustness and generalization.
   - In the similar manner, the validation of efficiency needs to be added to empirically prove that the proposed quantization is effective and efficient.
   - Also, the paper only deals with node classification tasks. It would be better if the paper included other graph-related tasks, such as node regression and graph classification, which have been actively dealt with in other graph Transformer papers.

**Questions:**

Please refer to the weaknesses.

---

> ### Author Response · Authors · 2024-11-22
> **Response to Reviewer rTiq**
>
> Thank you for the feedback and questions. Please find our responses as follows:
>
> **Novelty**
>
> We would like to clarify that our goal is not to introduce a new graph self-supervised objective or a novel quantization layer, as each of these topics warrants its own independent paper. Instead, our goal is to investigate an effective and transferable graph tokenization strategy that enables the learned tokens to encode local interactions, allowing a Transformer encoder to focus on long-range dependencies. To achieve this, we propose using state-of-the-art graph self-supervised objectives in a multitask fashion to capture various aspects of the interactions (as shown in Table 4, each objective contributes to the overall performance). Additionally, we have conducted new experiments showing that our method is robust and achieves strong performance across a diverse set of graph tasks. We believe our proposed approach will have a significant impact on the graph learning community by enabling researchers to leverage the accelerating advancements from the NLP and CV communities in designing more efficient Transformers.
>
> **Robustness and Generalization**
>
> Please refer to our new experiments discussed in the **[general response](https://openreview.net/forum?id=oYSsbY3G4o&noteId=Au8CNrOBD3)** to this question. In summary, we have performed new inference time attacks to evaluate robustness and conducted comparisons with RQ-VAE to assess generalization.
>
> **Efficiency**
>
> Please refer to our new experiments discussed in the **[general response](https://openreview.net/forum?id=oYSsbY3G4o&noteId=Au8CNrOBD3)** to this question.
>
> **Other Graph Tasks**
>
> Please refer to our new experiments discussed in the **[general response](https://openreview.net/forum?id=oYSsbY3G4o&noteId=Au8CNrOBD3)**  to this question. In summary, we have conducted new experiments on a diverse set of graph tasks including graph classification and regression, inductive node classification, and inductive link prediction on long-range graph benchmarks demonstrating the strong performance of our model.

---

> > ### Author Response · Authors · 2024-11-26
> > **Follow up**
> >
> > Hi reviewer rTiq,
> >
> > We tried our best to answer all of your questions and considering that we are approaching the end of rebuttal period, we would love to have more discussions with you and answer any further questions you may have.
> >
> > Thanks

---

> ### Author Response · Authors · 2024-12-01
> **Gentel Reminder**
>
> Dear Reviewer rTiq,
>
> Thank you again for your constructive comments. As the rebuttal period is coming to an end in one day, we would like to take this opportunity to kindly ask if you have any further questions based on our responses. We are more than happy to answer any additional questions during your re-evaluation.

---

> > ### Comment · Reviewer_rTiq · 2024-12-03
> >
> > Thank you for the detailed response.
> >
> > I have read all the reviews and the corresponding responses. I will increase my score to 5.

---

### Official Review · Reviewer_cTGj · 2024-11-08

**Soundness:** 3
**Presentation:** 3
**Contribution:** 3
**Rating:** 8
**Confidence:** 3

**Summary:**

the paper proposes a graph tokenizer that utilizes multi-task graph self-supervised objectives to train a graph encoder, which can capture local interaction and have memory efficiency

**Strengths:**

1. paper is well written and easy to follow
2. experiment demonstrate strong results compared with baseline methods.
3. motivation and methodology is clear

**Weaknesses:**

NO

**Questions:**

NO

---

> ### Author Response · Authors · 2024-11-22
> **Response to Reviewer cTGj**
>
> Thank you very much for your positive feedback. We would love to address any further questions you may have or clarify any part that is unclear to increase your confidence in your evaluation and in our work. Thanks again and looking forward to having a constructive discussion with you during the rebuttal period.

---

### Author Response · Authors · 2024-11-22
**General Response to AC and Reviewers - Part 1**

We extend our gratitude to the reviewers for their constructive feedback and insightful comments. We are pleased that the reviewers recognized the interest and relevance of our work. We acknowledge the common concerns raised by the reviewers and would like to address them in this general response. Additionally, we have updated the paper based on the feedback. All changes are highlighted in **red** in the updated manuscript.

**Performance on diverse graph tasks (reviewers rTiq, cy7L, D7qL)**

We would like to thank the reviewers for this great suggestion, which we believe will significantly improve the impact and quality of our paper. We initially focused on **transductive node classification** tasks, as many graph Transformer baselines [1-3] focus on these tasks. To further evaluate our model, we assessed its performance on graph classification, graph regression, link prediction, inductive node classification, and long-range modeling benchmarks from the **Long Range Graph Benchmark (LRGB)** [4], which is proposed to evaluate the capability of GNNs and GTs to capture long-range interactions. We used the Peptides-Func dataset for **graph classification** with Average Precision (AP) metric, the Peptides-Struct dataset for **graph regression** with Mean Absolute Error (MAE) metric (**lower is better**), the COCO-SP for **inductive node classification** with macro F1 metric, and the PCQM-Contact for **link prediction** with Mean Reciprocal Rank (MRR) metric. We compare our results to baselines reported in [5]. The results presented in the table below suggest that GQT coupled with the Transformer encoder can achieve promising results on all these tasks. In the camera-ready version, we will provide more comprehensive results by reporting our model’s performance on more datasets for each task. Notably, we did not modify the tokenizer and made only slight adjustments to the Transformer encoder, as discussed in Lines 149-152.

| Task | **Graph classification** | **Graph regression** | **Inductive Node classification** | **Inductive Link prediction** |
|---|---|---|---|---|
| Dataset | Peptides-Func | Peptides-Struct | COCO-SP | PCQM-Contact |
|Metric | AP ↑ | MAE ↓ | F1 ↑ | MRR ↑ |
| GCN | 0.5930±0.0023 | 0.3496±0.0013 |  0.0841±0.0010 | 0.3234±0.0006 |
| Exphormer | 0.6258±0.0092 | 0.2512±0.0025 | 0.3430±0.0108 | 0.3587± 0.0025 |
| GPS | 0.6535 ± 0.0041 | 0.2500 ± 0.0005 | 0.3412 ± 0.0044 | 0.3337 ± 0.0006 |
| Graph-Mamba | 0.6739±0.0087 | 0.2478±0.0016 | 0.3960±0.0175 | 0.3395± 0.0013 |
| Ours | 0.6903±0.0085 | 0.2452±0.0018 | 0.4007±0.0125 | 0.3427± 0.0012 |

**Robustness and Generalization (reviewers rTiq and cy7L)**

We would like to thank the reviewers for this great suggestion. To measure robustness, we use **PRBCDA and GRBCD inference time attacks** [6] to assess the performance drop during inference. A **lower drop in performance indicates greater robustness**. To this end, we compare the encoder of our tokenizer with the encoder of a tokenizer trained with a reconstruction objective (RQ-VAE) [7]. The results shown below suggest that our proposed tokenizer is more robust to attacks due to the multi-task graph-specific SSL objectives, as opposed to the standard RQ-VAE, which uses a reconstruction objective. In camera-ready version, we will provide additional results on other datasets.

| | GQT (ours) | RQ-VAE |
|---|---|---|
| GR-BCD Attack: Accuracy drop (PubMed) ↓     | 15.8% | 20.4% |
| GR-BCD Attack:Accuracy drop (ogbn-arxiv) ↓ | 10.4% | 14.8% |
| PR-BCD Attack: Accuracy drop (PubMed) ↓     | 18.1% | 23.3% |
| PR-BCD Attack: Accuracy drop (ogbn-arxiv) ↓ | 11.3% | 17.2% |

To measure improved generalization, we follow the common practice of treating **downstream predictive performance as a proxy for generalization**. As shown in Table 4 of the paper, every component of the tokenizer, including both SSL objectives and the quantization layer, contributes to the downstream predictive performance, thereby improving the model’s generalizability. Furthermore, to evaluate the contribution of multi-task SSL objectives to downstream performance, we compare our results with those of a tokenizer trained based on the RQ-VAE [7] design, which employs a reconstruction objective. The results presented below indicate that using multi-task SSL objectives significantly improves downstream predictive performance, which is strongly correlated with the method's generalization. In the camera-ready version, we will provide additional results on other datasets.

| Tokenizer | GQT (ours) | RQ-VAE |
|---|---|---|
| ogbn-arxiv | 73.14±0.16 | 66.05 ± 0.48 |
| Minesweeper | 95.28±0.44 | 89.69 ± 0.35 |

---

> ### Author Response · Authors · 2024-11-22
> **General Response to AC and Reviewers - Part 2**
>
> **Efficiency (reviewers rTiq and cy7L)**
>
> To address concerns about memory footprint, we have already highlighted two examples in Lines 483-485 and 426-429, showing significant memory reduction when using discrete tokens instead of node features. For instance, on the ogbn-products dataset with 2,449,029 nodes and 100-dimensional node features, GQT requires only 3 codebooks of size 4096 each to represent the tokens, resulting in a remarkable 30-fold reduction in memory usage. Please note that this memory reduction occurs after training the tokenizer. As the encoder of the tokenizer is a GNN that processes the graph with original node features, its memory footprint is on par with any arbitrary GNN. However, since the Transformer encoder only consumes discrete token IDs, which are significantly fewer than the total number of nodes, we achieve a substantial reduction in memory footprint. In the camera-ready version, we will include a table in the appendix to illustrate the improvement for all datasets. As an additional experiment, we compare the inference time and memory usage between our Transformer encoder and a Graph Attention Network (GAT) when performing inference on all graph nodes. The results below show that while our Transformer is on par with a sparse implementation of GAT in terms of inference time, it requires half the GPU memory.
>
> |  | ogbn-arxiv | Minesweeper |
> |---|---|---|
> | GAT | 2715MB (GPU Mem) / 5s (full inference) | 2108M (GPU Mem) / 1s (full inference)|
> | Ours | 1324MB (GPU Mem) / 4s  (full inference) | 1037MB (GPU Mem) / 1s  (full inference) |
>
> **References**
>
> [1] Fu et al. VCR-Graphormer: A Mini-batch Graph Transformer via Virtual Connections. ICLR 2024
>
> [2] Chen et al. NAGphormer: A Tokenized Graph Transformer for Node Classification in Large Graphs. ICLR 2023
>
> [3] Wu et al. NodeFormer: A Scalable Graph Structure Learning Transformer for Node Classification. NeurIPS 2022
>
> [4] Dwivedi et al. Long Range Graph Benchmark. NeurIPS 2022, Datasets and Benchmarks Track
>
> [5] Wang et al. Graph-Mamba: Towards Long-Range Graph Sequence Modeling with Selective State Spaces. Arxiv 2024
>
> [6] Geisler et al. Robustness of Graph Neural Networks at Scale. NeurIPS 2021
>
> [7] ​​Lee et al. Autoregressive Image Generation using Residual Quantization. CVPR 2022

---

### Meta-Review · Area_Chair_dtA8 · 2024-12-14

**Metareview:**

**Summary:**

The paper proposes a self-supervised method for graph token learning, Graph Quantized Tokenizer (GQT). The proposed framework requires two-stage training; the tokenizer and the model are separately trained. Specifically, the authors adopt Residual Vector Quantization to learn hierarchical discrete tokens. Extensive experiments demonstrate that the proposed framework improves memory efficiency and generalizability, achieving superior performance across various datasets including large-scale datasets.

**Strengths:**

1. **Strong empirical results and versatility.** The authors demonstrate the superiority of their model across various transductive node classification datasets, including both homophilic and heterophilic graphs, and large-scale graphs with 2.5 million nodes. Also additional experimental results on graph-level, and link-level tasks support the versatility of the proposed method.
2. **Memory-efficient graph transformer.** The proposed Graph Quantized Tokenizer utilizes discrete tokens, which can significantly reduce memory requirements.
3. **Motivation.** Designing and learning tokenizers in graph Transformers are important research topics. The authors proposed a learnable tokenizer for graphs using vector quantization. Decoupling the training of the graph tokenizer and graph transformer is interesting to further explore.
4. **Presentation.** Paper is well-written and provides comprehensive related works.

**Weaknesses:**

1. **Limited technical novelty.** This paper successfully combines existing modules, but the modules are not newly developed in this paper. The proposed pipeline can be viewed as a simple combination of existing graph Transformers and residual vector quantization. Also loss functions for self-supervised learning have already been studied in the literature.

**Main reasons:**

This paper studies a graph transformer equipped with a learnable tokenizer to improve performance and enable large-scale graph analysis. Although the modules are mainly from existing works in the literature, the proposed pipeline is well-motivated and shows strong performance. Also, the authors addressed most concerns raised by reviewers via responses with empirical results.

**Additional Comments On Reviewer Discussion:**

Most concerns raised by reviewers are addressed by the authors' responses. Two main concerns are addressed as the following.

1. **More tasks.** The authors provided graph classification, graph regression, inductive node classification, and inductive link prediction on long-range benchmarks.
2. **Memory efficiency.** The authors provide an additional table to show the memory efficiency of the proposed method.

---

### Decision · Program_Chairs · 2025-01-22

Accept (Poster)